# Rethinking Alignment in Video Super-Resolution Transformers

**Shuwei Shi**[1,2,*]**, Jinjin Gu**[3,4,*]**, Liangbin Xie**[2,5,6]**, Xintao Wang**[6]**, Yujiu Yang**[1]**, Chao Dong**[2,3,†]

[1] Shenzhen International Graduate School, Tsinghua University
[2] Shenzhen Institutes of Advanced Technology, Chinese Academy of Sciences
[3] Shanghai AI Laboratory      [4] The University of Sydney
[5] University of Chinese Academy of Sciences      [6] ARC Lab, Tencent PCG
`jinjin.gu@sydney.edu.au, ssw20@mails.tsinghua.edu.cn,`
`{chao.dong, lb.xie}@siat.ac.cn, xintaowang@tencent.com`
`yang.yujiu@sz.tsinghua.edu.cn`

## Abstract

The alignment of adjacent frames is considered an essential operation in video super-resolution (VSR). Advanced VSR models, including the latest VSR Transformers, are generally equipped with well-designed alignment modules. However, the progress of the self-attention mechanism may violate this common sense. In this paper, we rethink the role of alignment in VSR Transformers and make several counter-intuitive observations. Our experiments show that: (i) VSR Transformers can directly utilize multi-frame information from unaligned videos, and (ii) existing alignment methods are sometimes harmful to VSR Transformers. These observations indicate that we can further improve the performance of VSR Transformers simply by removing the alignment module and adopting a larger attention window. Nevertheless, such designs will dramatically increase the computational burden, and cannot deal with large motions. Therefore, we propose a new and efficient alignment method called patch alignment, which aligns image patches instead of pixels. VSR Transformers equipped with patch alignment could demonstrate state-of-the-art performance on multiple benchmarks. Our work provides valuable insights on how multi-frame information is used in VSR and how to select alignment methods for different networks/datasets. Codes and models will be released at `https://github.com/XPixelGroup/RethinkVSRAlignment`.

## 1  Introduction

Video Super-Resolution (VSR) could provide more accurate SR results than Single-Image Super-Resolution (SISR) by exploiting the complementary sub-pixel information from multiple frames. The well-established paradigms of VSR networks usually include an alignment module to compensate for the motion of objects between frames. The alignment module is critical for CNN-based VSR networks, because the locality inductive bias of CNNs only allows them to utilize spatial-close distributed information effectively. Many VSR networks have achieved better performance by introducing more advanced alignment methods [43, 4, 39, 24].

Recently, the general paradigm of vision network design has gradually shifted from CNNs to Transformers [10, 23, 7]. Unlike the locality property of CNNs, the self-attention operation in Transformers is very efficient for processing elements with spatially long-term distribution. VSR is no exception: Transformers have also been introduced into VSR for better results [3, 22, 24]. However,

---

[*]Jinjin Gu and Shuwei Shi contribute equally to this work.
[†]Corresponding author.

36th Conference on Neural Information Processing Systems (NeurIPS 2022).

these VSR Transformers still retain complex alignment modules. The ability of Transformers to efficiently process non-local information has not yet been exploited for inter-frame information processing.

In this paper, we rethink the role of the alignment module in VSR Transformers. We make two arguments: **(i)** *VSR Transformers can directly utilize multi-frame information from unaligned videos*, and **(ii)** *continuing to use the existing alignment methods will sometimes degrade the performance of VSR Transformers*. For the first argument, we report that for Transformers using the shifted window mechanism [30, 23], misalignment within a certain range does not affect the SR performance, while alignment only has a positive effect on the pixels beyond that range. The attribution map visualization results [11] also show that Transformers without alignment modules behave similarly to CNNs with alignment modules. For the second argument, we quantitatively compare the effects of various alignment methods. Using alignment will only have negative effects for pixels without large misalignment (*e.g.*, Vimeo-90K [47]). We blame this phenomenon on the noise of optical flow and the destruction of sub-pixel information by alignment resampling operations. Although implicit alignment based on deformable convolution can minimize these negative effects, the additional cost of parameters and computation makes this approach no longer advantageous.

According to our findings, we can build an alignment-free VSR Transformer, which requires a large attention window (>16) to cover misalignment between frames. However, enlarging the window size will lead to higher computational costs, making this approach no longer feasible. As a better alternative, we propose a simple yet effective alignment method called Patch Alignment, which aligns image patches instead of pixels. It will find the corresponding patches in the supporting frames and compute self-attention among them. This method uses a simple crop-then-move strategy to compensate for situations where the misalignment exceeds the Transformer's attention window. Experiments show that the proposed method achieves state-of-the-art VSR performance with a simple design and fewer parameters.

## 2  Related Works

**VSR networks.**  The uniqueness of the VSR task lies in the utilization of inter-frame sub-pixel information [29]. Due to the motion of objects between frames, the computation and compensation for this motion is often the focus of VSR research. The pipeline of most VSR networks mainly includes an alignment module, a feature extraction module, a fusion module, and a reconstruction module. These networks are trained using a large number of low-resolution (LR) and high-resolution (HR) video sequences. The paradigm of early methods [25, 15, 28, 16, 47, 14, 26] can be summarized as: first estimate and compensate object motion (using optical flow), then pass through different feature extraction [36, 46, 19], fusion [38, 2, 4], and reconstruction modules [37], respectively. These methods can be further divided into methods for compensating motion in the image space (image alignment) and that in the feature space (feature alignment) [4, 42]. As Chan *et al.*[4] claims, feature alignment can achieve better results because inaccuracy in flow estimation may be mitigated after feature extraction.

However, the above methods generally suffer from inaccurate optical flow estimation and defective alignment operation, resulting in noticeable artifacts in the aligned images [39]. With the invention of Deformable convolutions [9], CNN networks also have the ability to model geometric transformations. Deformable convolutions gradually replace explicit warping operations in VSR networks with their learnable and flexible properties [43, 51, 39]. EDVR [43] integrates deformable convolution in the VSR network for the first time and aligns features at multiple scales. Chan *et al.*[5] incorporate optical flow as a guidance for offsets learning. The latest BasicVSR++ [6] based on the bidirectional loop structure also uses the alignment based on deformable convolution.

Differently, our paper aims to study Transformers' ability to model multi-frame information implicitly without using alignment. Before Transformer, attempts have been made to achieve the same effect in CNN, *i.e.*, the VSR methods without alignment. Both 2D [32, 48] and 3D convolutions [40, 13, 20, 16] are used to extract correlations among frames. Similar to Transformers, some methods use correspondence calculation to build inter-frame connections without alignment. For example, MuCAN [21] uses a temporal multi-correspondence aggregation module and a cross-scale non-local-correspondence aggregation module to replace explicit alignment. Yi *et al.*[50] introduce an improved non-local operation to avoid the complex alignment procedure. Generally speaking, models without explicit alignment are less effective or rely on special module design. However, the Transformers discussed in this paper could efficiently handle unaligned frames without additional special design.

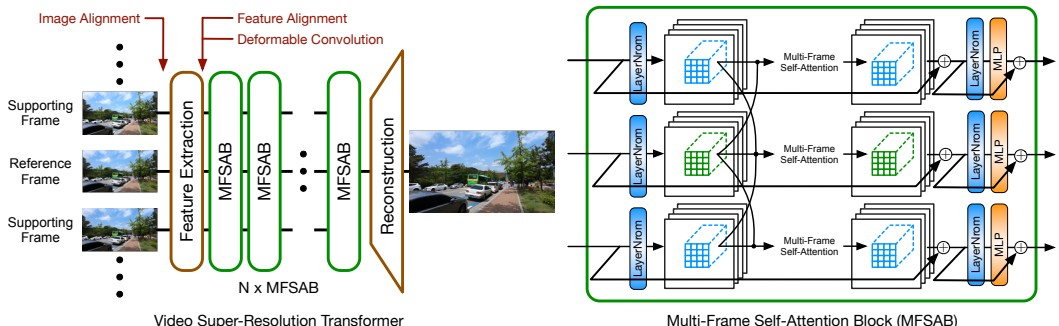

Figure 1: A brief illustration of the VSR transformer network used. The illustrated structure is based on the sliding window mechanism. We mark the position of the existing alignments methods.

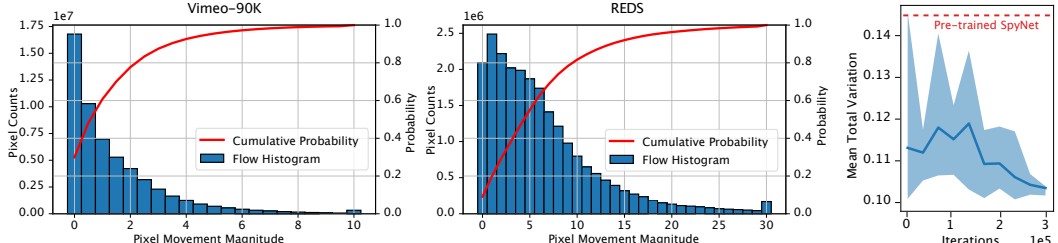

Figure 2: The distribution of the movement for the Vimeo-90K [47] and REDS [33] test sets. The distribution of pixel movement exhibits a long-tailed feature. The REDS dataset contains a larger movement.

Figure 3: The curve of total variation of the fine-tuned flow.

**Transformer networks.** Transformers [41] are a new type of networks characterized by self-attention operations. A Transformer is considered to have a parameter-independent global receptive field within the scope of self-attention. Dosovitskiy *et al.*[10] first introduces Transformer into image recognition by projecting large image patches into a sequence of tokens. Inspired by the success of vision Transformers, many attempts have been made to employ Transformers for low-level vision tasks [49, 52, 7, 45, 53, 8] As a representative model, Liang *et al.*proposes SwinIR [23] that integrates a powerful shifted window strategy and achieves promising results. Transformer-based models have also been proposed for VSR. Cao *et al.*[3] employ the spatial-temporal self-attention on multi-frame patches to perform implicit motion compensation. Liang *et al.*[22] propose a parallel frame prediction Transformer to model long-range temporal dependency. Both the above Transformers consist of alignment modules. VSRT [3] uses feature alignment method and VRT [22] employs complex deformable convolution to perform forward and backward alignment. In this paper, we show that these complex alignment operations are unnecessary or even harmful for VSR Transformers.

## 3 Preliminary Settings

In this section, we describe the settings used in our experiments in detail, including the Transformer architecture, alignment methods, datasets, metrics and implementation details. If the readers are already familiar with the common VSR settings and VSR networks, it is okay to skip this section and go to Section 4 for experimental results and analysis. Due to space constraints, we record the detailed setup of the different experiments in Appendix **??** to facilitate the readers to reproduce our results.

**The VSR Transformer architecture.** We first describe the basic VSR Transformer backbone used in this study, which follows the sliding window design. Transformer based on shifted window mechanism [23] is proven to be flexible and effective for image processing tasks. We only make minimal changes when applying it to VSR to maintain its good performance without the loss of generality. This model is illustrated in Figure 1. The VSR Transformer takes a reference frame $I^t$ and $2n$ adjacent supporting frames $\{I^{t-n}, \ldots, I^{t-1}, I^{t+1}, \ldots, I^{t+n}\}$ as input, and outputs the super-resolved reference frame $I_{SR}^t$. In the beginning, a feature extraction module is used to extract features for the subsequent Transformer. We use a single 2D convolution layer for feature extraction. The extracted features are denoted as $2n + 1$ feature maps $\{X^{t-n}, \ldots, X^t, \ldots, X^{t+n}\}$.

Then $N$ Multi-Frame Self-Attention Blocks (MFSAB) are used as the backbone network. The MFSAB is modified from RSTB in SwinIR [23], which contains two LayerNorm [1] layers, a multi-head self-attention layer and a multi-layer perceptron layer. We mainly modify the self-

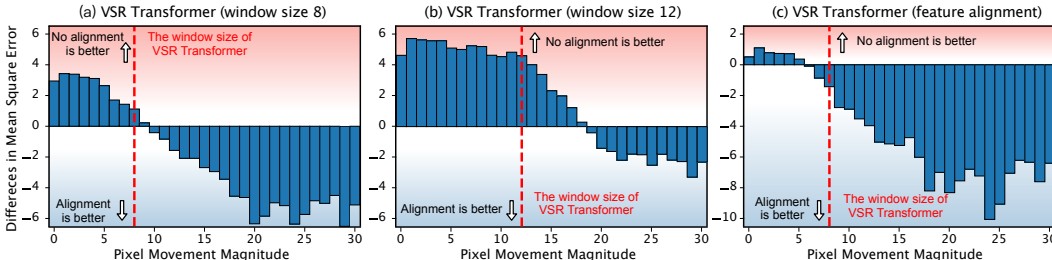

Figure 4: This figure illustrates the performance differences between VSR Transformers with and without alignment module for different pixel movement conditions. Parts greater than zero indicate better performance without alignment.

attention layer to make it suitable for the VSR task. Given these $2n + 1$ feature maps of the frames with size $H \times W \times C$ ($H$, $W$ and $C$ are the height, width and feature dimension), the shifted window mechanism first reshapes the feature maps of each frame to $\frac{HW}{M^2} \times M^2 \times C$ features by partitioning the input into non-overlapping $M \times M$ local windows, where $\frac{HW}{M^2}$ is the total number of windows. We calculate self-attention on the features in the windows corresponding to the positions in different frames. Therefore, $(2n + 1)M^2$ features are involved in each standard self-attention operation, and we concatenate features from different frames to produce a local window feature $X \in \mathbb{R}^{(2n+1)M^2 \times C}$. In each self-attention layer, the query $Q$, key $K$ and value $V$ are computed as $Q = XW^Q, K = XW^K, V = XW^V$, where $W^Q, W^K, W^V \in \mathbb{R}^{C \times D}$ are weight matrices, and $D$ is the channel number of projected vectors. Then, we use $Q$ to query $K$ to generate the attention map $A = \texttt{softmax}(QK^T/\sqrt{D} + B) \in \mathbb{R}^{(2n+1)M^2 \times (2n+1)M^2}$, where $B$ is the learnable relative positional encoding. This attention map $A$ is then used for the weighted sum of $(2n + 1)M^2$ vectors in $V$. The multi-head settings are aligned with SwinIR [23] and ViT [10]. In the final reconstruction module, only the features of the reference frame are used to generate the SR result. We use the pixel-shuffle layer [37] to upsample the feature maps. Besides sliding window based VSR Transformer, the MFSAB can also be applied to recurrent frameworks [6]. We describe the details in Appendix **??**.

**Alignment methods.** Alignment methods follow a long line of research. To investigate the role of alignment, we need to implement and compare different alignment methods. These alignment methods can be classified into four types, and their respective representative methods are included in our experiments. We next describe their characteristics and then describe our implementation.

- *Image alignment* is the earliest and most intuitive alignment method, which is also the easiest to use. Image alignment relies on the explicitly calculated optical flow between frames. According to the estimated inter-frame motion, different frames are aligned by a warping operation. Following the existing successful experience [47, 4], we use the SpyNet [35] to estimate the optical flow, and fine-tune the SpyNet simultaneously during training. The resampling method used is the bilinear (BI) method.

- *Feature alignment* also estimates the optical flow but performs the warping operation on the deep features instead of on the images. The flow estimation module still uses SpyNet, which is optimized during training. In addition to the shallow 2D convolution in Figure 1, we add five additional residual blocks [12] to extract deep features.

- *Deformable convolution (DC) based methods* use learnable dynamic deformable convolution for alignment. Almost all state-of-the-art VSR networks use deformable convolution to perform the alignment. We employ the flow-guided deformable convolution (FGDC) alignment used in BasicVSR++ [6] and VRT [22] as the representative method.

- *No alignment.* The raw input is processed directly using the VSR Transformer.

**Datasets and metrics.** In the VSR literature, the REDS [33] and the Vimeo-90K [47] datasets are the de-facto benchmarks. REDS has 270 available video sequences, each containing 100 frames. We follow the common splitting methods and split the data into training (266 sequences) and testing (4 sequences) sets. Vimeo-90K contains 64,612 and 7,824 video sequences for training and testing, respectively. Although REDS and Vimeo-90K are widely used benchmarks, these two datasets have different motion conditions. The motion in the Vimeo-90K dataset is generally small, and movement magnitudes of 99% pixels are less than 10 (for each clip, we measure the motion of the 4th and the 7th frames). Differently, there are large motions in the REDS dataset. There are at least 20% pixels that have movement magnitudes larger than 10 (for each clip, we measure the motion of the 3rd and the 5th frames). The distributions of the movement conditions for these two datasets are shown in

Table 1: Quantitative comparison of different VSR methods. The results marked with ∗ achieve similar performance as no alignment. This is due to the vanishing of optical flow in this experiment. Details are shown in Figure 5 and discussed in Section 4.2

| Exp. Index | Method | Alignment | Remark | Vimeo90K-T PSNR | Vimeo90K-T SSIM | REDS4 PSNR | REDS4 SSIM |
|---|---|---|---|---|---|---|---|
| 1 | VSR-CNN | Image alignment | Finetune flow | 36.13 | 0.9342 | 29.81 | 0.8541 |
| 2 | VSR-CNN | No alignment | | 36.24 | 0.9359 | 28.95 | 0.8280 |
| 3 | VSR Transformer | Image alignment | Fix flow | 36.87 | 0.9429 | 30.25 | 0.8637 |
| 4 | VSR Transformer | Image alignment | Finetune flow | 37.44∗ | 0.9472∗ | 30.43 | 0.8677 |
| 5 | VSR Transformer | Feature alignment | Finetune flow | 37.36 | 0.9468 | 30.74 | 0.8740 |
| 6 | VSR Transformer | No alignment | Window size 8 | 37.43 | 0.9470 | 30.56 | 0.8696 |
| 7 | VSR Transformer | No alignment | Window size 16 | 37.46 | 0.9474 | 30.81 | 0.8745 |

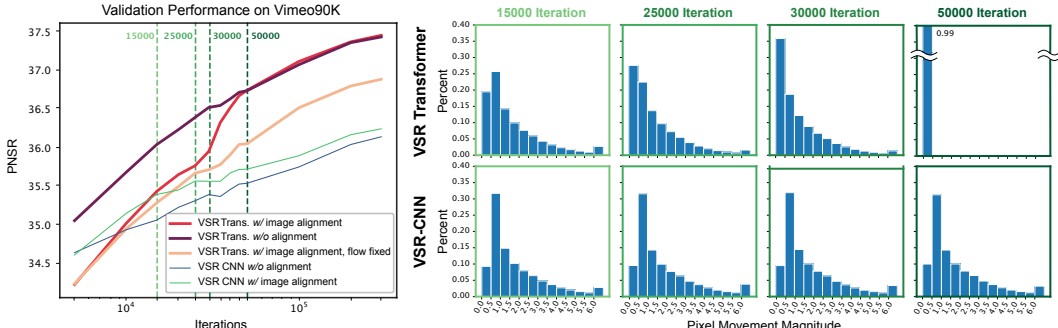

Figure 5: The figure on the left shows the validation curves tested on Vimeo-90K test set. On the right are the histogram distribution of the estimated optical flow at different training iterations. The moment when the optical flow distribution changes corresponds to the moment when the performance of the VSR Transformer with alignment increases.

Figure 2. In our experiments, we mainly study ×4 VSR task. We use bicubic interpolation to produce LR video frames. Peak signal-to-noise ratio (PSNR) and structural similarity index (SSIM) are used for evaluation.

**Implementation.** We implement all these methods using the BasicSR framework [44]. We use the Charbonnier loss [18] as the training objective. The Adam optimization [17] method is used for training with $\beta_1 = 0.9$ and $\beta_2 = 0.999$. The initial learning rate is set to $4 \times 10^{-4}$, and a Consine Annealing scheme [31] is used to decay the learning rate. The total iteration number is set to 300,000. The mini-batch size is 8, and the LR patch size is $64 \times 64$. The experiments are implemented based on PyTorch [34] and conducted on NVIDIA A100 GPUs.

## 4 Rethinking Alignment

In this section, we conduct experiments based on the above preliminary settings. We use four questions to guide our rethinking. We present our experimental results and analysis in each subsection.

### 4.1 Does alignment always benefit VSR Transformers?

In previous VSR research, we often judge the pros and cons of the method based on the average performance on the test set. But for the VSR Transformers, the performance of alignment inside the local window is different from that outside the window because of the limited range of self-attention. A finer perspective on how different methods perform on different data may help us understand the effect of alignment. We investigate the performance difference between the VSR Transformer with and without alignment for different pixel movement conditions. Figure 4 shows the results tested on the REDS dataset and Table 1 reports the quantitative results for these experiments.

One can draw the following observations. First, as can be observed from Figure 4 (a), for pixels with small movements, VSR Transformer can achieve good results *without* alignment. Using image alignment in these cases brings negative effects. This range of pixel movement is related to the window size used by the VSR Transformer. As there is no locality inductive bias in the processing of pixels within a local window, the Transformer can handle misalignment in this range. Second, as the movement increases, the information required by VSR exceeds the scope of the local windows. In these cases, image alignment can improve performance. Third, according to Figure 2, the movement

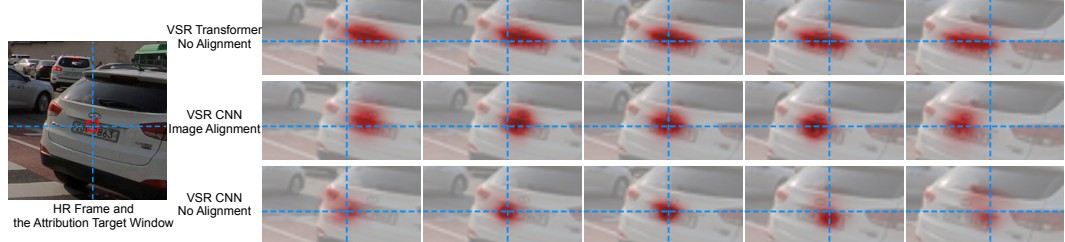

Figure 6: The attribution results of different models. We use the local attribution map [11] to highlight the responsible pixels in each frame for the SR of the selected red window.

Table 2: Comparison of VSR Transformers with different alignment methods on the REDS4 dataset.

| # | \multicolumn{4}{c}{Alignment Method} | | | | \multicolumn{2}{c}{Position} | \multicolumn{2}{c}{Resampling} | Params. | REDS4 |
| | No Ali. | Img. Ali. | Feat. Ali. | FGDC | Img. | Feat. | BI | NN | (M) | PSNR / SSIM |
|---|---|---|---|---|---|---|---|---|---|---|
| 1 | ✓ | | | | | | | | 12.9 | 30.92 / 0.8759 |
| 2 | | ✓ | | | ✓ | | ✓ | | 12.9 | 30.84 / 0.8752 |
| 3 | | | ✓ | | | ✓ | ✓ | | 14.8 | 31.06 / 0.8792 |
| 4 | | | ✓ | | | ✓ | | ✓ | 14.8 | 31.11 / 0.8801 |
| 5 | | | | ✓ | | ✓ | | | 16.1 | 31.11 / 0.8804 |

of about 70% pixels is less than eight on the REDS test set. This makes the model without alignment still perform better than the image alignment on this test set.

We then increase the size of the shifted window, and conduct the same experiment. The result is shown in Figure 4 (b). We can see that VSR Transformer can handle a larger range of unaligned pixels with a larger window size. This shows that the processing capability of VSR Transformers for unaligned frames is related to the size of the shifted window, and also implies that this capability mostly relies on the self-attention mechanism. To investigate whether better alignment methods can eliminate the negative effects of alignment on small motion pixels, we use feature alignment to conduct the same experiment, and the result is shown in Figure 4 (c). As one can see, feature alignment narrows the gap between alignment and no alignment on VSR Transformer, but it still has negative effects on the pixels of small motion.

In summary, the VSR Transformer can handle misalignment within a certain range, and using alignment at this range will bring negative effects. This range is closely related to the window size of the VSR Transformer. However, alignment is necessary for motions beyond the VSR Transformer's processing range.

## 4.2 What kinds of flow are better for VSR?

Although using optical flow for alignment can have a negative impact, different flows can also lead to differences in performance. This inspires us to think about what kinds of flow are better for VSR? According to Table 1, one can see that optimizing the flow estimator while training the VSR network will bring better results, because the flow estimator at this time learns the optimized flow for VSR [47]. The property of this task-oriented flow can imply how the VSR Transformers use multi-frame information. Our first observation is that VSR Transformers tend to use smooth flow. The flow estimator SpyNet was pre-trained with the end-point-error (EPE) loss, which does not explicitly encourage smoothness. The non-smooth flow will introduce random noise to VSR and lose sub-pixel information. To study this problem, we investigate the change of the estimated flow while training the VSR Transformer with image alignment using the REDS dataset. We can see that the flow estimated by the fine-tuned SpyNet is getting smoother, which is reflected in the decrease of the average total variation, as shown in Figure 3. Smoother flow maintains the relative relationship of adjacent pixels in the aligned frames, thus facilitating VSR processing.

Although the fine-tuned flow estimator will improve performance, there is still a gap between image alignment with flow fine-tuning and no alignment on the REDS dataset. However, we observe different results on the Vimeo-90K dataset: image alignment with flow fine-tuning is almost identical to no alignment. By observing the distribution of the estimated flow, we are surprised to find that the flow is slowly decreasing to 0 when fine-tuned with image alignment using Vimeo-90K. This is why the final result is almost the same as no alignment. In Figure 5, we show the validation curve of the related experiments and the histogram distribution of the fine-tuned flow. As one can see, there is a large gap between the VSR Transformers with and without alignment in the early stage of training. When the training reaches 25,000 iterations, the fine-tuned flow gradually shifts towards the zero

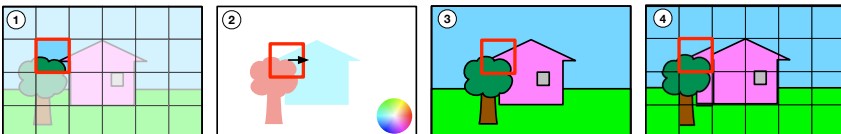

Figure 7: The pipeline of the proposed Patch Alignment method: ① partition the input frames to patches according to the window partition of Transformer, ② calculate the mean motion vector for each patch, ③ find the corresponding patches in the supporting frames, and ④ move the entire supporting patches to their corresponding position.

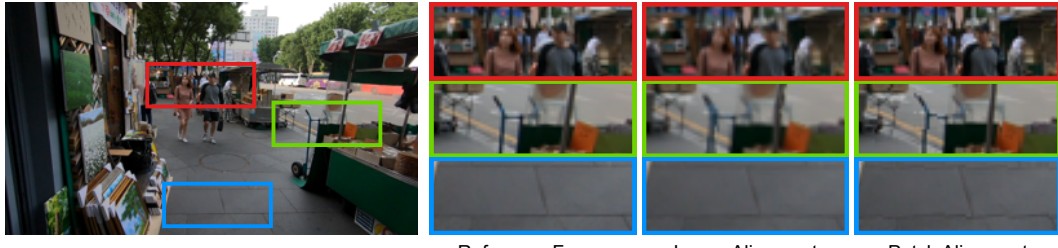

Reference Frame  Image Alignment  Patch Alignment

Figure 8: The visualization comparison of image alignment and the proposed patch alignment.

direction. When training after 50,000 iterations, the fine-tuned flow vanishes. At this time, the VSR Transformer with alignment is equivalent to the one without alignment. This phenomenon does not appear on VSR-CNN. This experiment is instructive. On the one hand, most of the movements in the Vimeo-90K dataset are smaller than the Transformer's window size. According to Figure 4 (a), alignment is detrimental to training at this situation. The fine-tuned flow estimator seems aware of this knowledge and learns to improve performance by forcing flow value to all zeros. On the other hand, the adaptability of the model is surprising. The model will choose the policy that maximizes the training objective, even if that policy completely disables part of the network.

We can now answer the question posed by this subsection. For most VSR Transformers, the smooth flow is better if the alignment is necessary. For situations with small motions, the flow with all zeros is the best for VSR Transformers.

### 4.3  Does Transformer implicitly track the motion between unaligned frames?

We already know that VSR Transformers can handle a specific range of mis-alignment, but do they track these movements implicitly? Can an alignment-like function be done inside the VSR Transformers? We next use an inter-pretability tool to visualize the behaviour of the VSR Transformer. Local Attribution Map (LAM) [11] is an attribution method aiming to find input pixels strongly influencing network output. We first specify a target patch on the output image and then use LAM to generate the corresponding attribution maps. We will track the information used by the model, and see which parts of pixels in adjacent frames contribute the most. As objects move between frames, an ideal VSR network needs to track those movements and utilize pixels representing the same object. We show some representative results in Figure 6. It can be observed that even without the alignment module, the VSR Transformer can automatically change its attention to the most relevant pixels.

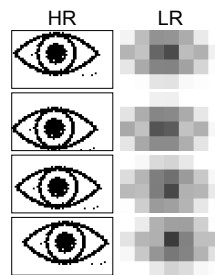

Figure 10: Illustration of sub-pixel information.

Intuitively, the self-attention operation can detect the associations between pixels in different frames and utilize their additional information for VSR. This is similar to VSR CNNs with the alignment module. We also show results of VSR CNNs without the alignment module. Since CNN has the inductive bias of locality, it can only focus on the area near the current position if no alignment is used. Thus, alignment is an essential module for CNN-based VSR methods.

### 4.4  Why do alignment methods have negative effects?

To understand the reasons for the negative impact of alignment, we need to know what sub-pixel information does VSR require. As shown in Figure 10, high-frequency information in HR frames is lost during downsampling, and only aliasing patterns are left in the LR frames. When the HR frames move, different LR frames are generated, and different aliasing patterns are produced. These patterns provide additional constraints for VSR. However, the inaccurate optical flow and the bilinear

| Method | Position | | Resampling | | REDS4 | |
|---|---|---|---|---|---|---|
| | Img. | Feat. | BI | NN | PSNR | SSIM |
| Patch Alignment | ✓ | | | ✓ | 31.11 | 0.8800 |
| | | ✓ | ✓ | | 31.00 | 0.8781 |
| | | ✓ | | ✓ | 31.17 | 0.8810 |

Table 3: The Ablation study of Patch Alignment. We study the effect of different resampling methods (BI and NN) and different alignment positions (image space and feature space).

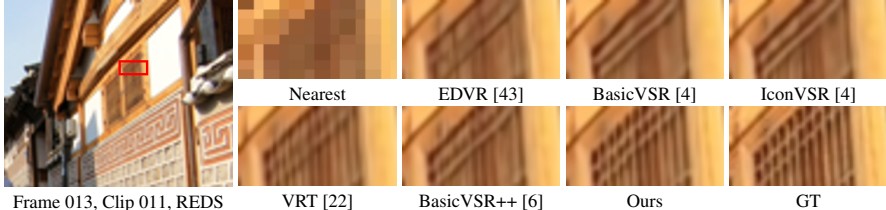

Nearest  EDVR [43]  BasicVSR [4]  IconVSR [4]

Frame 013, Clip 011, REDS  VRT [22]  BasicVSR++ [6]  Ours  GT

Figure 9: Visual comparison of VSR ($\times 4$) on REDS dataset.

resampling operation could corrupt these patterns. First, the inaccurate flow can be viewed as a combination of the ground-truth flow and a random error term. Using such flow for alignment randomly will change the LR patterns and cause information loss. Second, the bilinear resampling operation calculates the weighted average among four adjacent pixels, and the weights are inaccessible for the VSR model. At this time, the interpolation is irreversible. The VSR model can only process the transformed LR patterns and cannot access the original patterns, resulting in information loss. We conduct experiments to demonstrate the negative effects of optical flow and resampling methods. The results are shown in Table 2. Compared with image alignment, feature alignment improves performance by extracting part of the sub-pixel information before it is corrupted by alignment. The flow-guided deformable convolution (FGDC) reduces the negative effects of alignment by enabling the network to model geometric transformations. Changing the resampling method to nearest neighbor (NN) also improves performance because the NN method can preserve the relationship between adjacent pixels and ignore the noise of flow estimation to a certain extent. As can be seen, feature alignment using the NN resampling method achieves the same performance as the FGDC method but with a significantly reduced number of parameters.

## 5 Patch Alignment

According to our observations, an inaccurate flow and the resampling operation will impair the utilization of the inter-frame information. To overcome this problem, simply increasing the Transformer's window size is a straightforward solution as Transformers are naturally good at modelling unaligned spatial dependencies within the local window. However, the time and space complexity of increasing the window size is at least $\mathcal{O}(n^2)$. We need more efficient alignment methods to improve performance for large movement pixels while introducing little negative impact and additional computational complexity. In this section, we propose a simple yet effective alignment method for VSR Transformers, called Patch Alignment.

### 5.1 Method

The pipeline of the proposed method is shown in Figure 7. This method does not align individual pixels but treats the image as non-overlapping patches. The partition of patches is consistent with the partition of the Transformer's local windows. We treat the patch as a whole and perform the same operation on the pixels within the patch. In this way, the relative relationship between pixels is kept, and resampling operations will not corrupt the sub-pixel information within the patch. We locate the movement of objects based on optical flow, yet we do NOT pursue precise pixel-level alignment. We calculate the mean motion vector in each patch and find the corresponding patches in the supporting frames for each patch. Next, we use the nearest neighbor resampling method to move the entire supporting patches to their corresponding position in the reference frame. The benefits are three-fold. First, the nearest neighbor resampling ignores the fractional part of optical flow estimation and reduces the error caused by inaccurate flow estimation. Second, the entire patch is cropped and moved to the corresponding position, which preserves the relative relationship of the pixels within the patch and thus retains the sub-pixel information.

We show the comparison of image alignment with bilinear resampling and the proposed patch alignment method in Figure 8. As can be seen, image alignment introduces blurry and artifacts to the aligned image that destroy sub-pixel information. Patch alignment retains more details that can

Table 4: Quantitative comparison (PSNR↑ and SSIM↑) on the REDS4 [33] dataset, Vid4 [27], Vimeo-90K-T [47] dataset for $4\times$ VSR task. Red indicates the best and blue indicates the second best performance (best view in color) in each group of experiments.

| Method | Frames REDS/Vimeo | Params (M) | REDS4 PSNR | REDS4 SSIM | Vimeo-90K-T PSNR | Vimeo-90K-T SSIM | Vid4 PSNR | Vid4 SSIM |
|---|---|---|---|---|---|---|---|---|
| EDVR [43] | 5/7 | 20.6 | 31.09 | 0.8800 | 37.61 | 0.9489 | 27.35 | 0.8264 |
| VSR-T [3] | 5/7 | 32.6 | 31.19 | 0.8815 | 37.71 | 0.9494 | 27.36 | 0.8258 |
| PSRT-sliding | 5/- | 14.8 | 31.32 | 0.8834 | - | - | - | - |
| VRT | 6/- | 30.7 | 31.60 | 0.8888 | - | - | - | - |
| PSRT-recurrent | 6/- | 10.8 | 31.88 | 0.8964 | - | - | - | - |
| BasicVSR [4] | 15/14 | 6.3 | 31.42 | 0.8909 | 37.18 | 0.9450 | 27.24 | 0.8251 |
| IconVSR [4] | 15/14 | 8.7 | 31.67 | 0.8948 | 37.47 | 0.9476 | 27.39 | 0.8279 |
| BasicVSR++ [6] | 30/14 | 7.3 | 32.39 | 0.9069 | 37.79 | 0.9500 | 27.79 | 0.8400 |
| VRT | 16/7 | 35.6 | 32.19 | 0.9006 | 38.20 | 0.9530 | 27.93 | 0.8425 |
| PSRT-recurrent | 16/14 | 13.4 | 32.72 | 0.9106 | 38.27 | 0.9536 | 28.07 | 0.8485 |

provide additional information for the VSR model. As we do not pursue pixel-level alignment, directly operating on patches will leave discontinuous artifacts along patch borders. But our experiments show that these discontinuities have little effect on VSR Transformers. Because these discontinuities do not appear in the local window of the Transformer, they do not affect the function of self-attention. It further illustrates the importance of preserving sub-pixel information.

## 5.2 Experimental results

We test the patch alignment method with different alignment positions and resampling methods. The experimental settings and the network configurations are the same with experiments in Table 2. The results are shown in Table 3. As can be seen, image-level patch alignment is already comparable to FGDC, while the latter uses 25% more parameters. Feature-level patch alignment achieves the best performance among all the tested alignment methods. We conduct an ablation experiment using bilinear resampling to study the importance of NN resampling for patch alignment. It can be seen that bilinear resampling leads to a severe performance drop. It shows that the LR patterns retained by the NN method are critical for VSR.

We use patch alignment to build VSR Transformers based on the sliding window and the recurrent frameworks, namely PSRT-sliding and PSRT-recurrent. We test their performances on REDS and Vimeo-90K datasets. The results are shown in Table 4. VSR Transformers with patch alignment can achieve state-of-the-art performance with fewer parameters compared to other Transformer-based VSR methods (VSRT [3] and VRT [22]). Compared to BasicVSR++ [6], we use fewer frames in training, but achieve a 0.33dB improvement on REDS in terms of PSNR. Visual results of different methods are shown in Figure 9. As one can see, in accordance with its significant quantitative improvements, the proposed method can generate visually pleasing images with sharp edges and fine details. By contrast, its competitors suffer from either blurry or lost details.

## 6 Conclusion

In this paper, we present two important conclusions about using Transformers in the VSR task: (i) VSR Transformers can directly utilize multi-frame information from unaligned videos, and (ii) existing alignment methods are sometimes harmful to VSR Transformers. We also propose a new patch alignment method for VSR Transformers. The proposed method demonstrates the state-of-the-art performance for VSR.

Given the current literature, our results are interesting and inspiring. They challenge our common understanding of using Transformers to process multiple spatially misaligned images. First, the analysis of alignment can provide useful insights for VSR. We need to utilize inter-frame sub-pixel information, yet many image pre-process operations interfere with our utilization of this information. Second, these observations hint that Transformer can implicitly make accurate connections for misaligned pixels. Many low-level vision tasks can take advantage of this property, such as video restoration, reference-based SR, burst image processing, stereo matching, *etc*. When designing Transformers for these tasks, we can no longer explicitly introduce the alignment modules or the cost volume modules, but give full play to the powerful modeling capabilities of the Transformer itself.

**Acknowledgement** This work was supported in part by the National Natural Science Foundation of China (Grant No. 61991450) and the Shenzhen Key Laboratory of Marine IntelliSense and Computation (ZDSYS20200811142605016).

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
