# Supplementary Materials: Rethinking Alignment in Video Super-Resolution Transformers

**Shuwei Shi**[1,2,*]**, Jinjin Gu**[3,4,*] **, Liangbin Xie**[2,5,6]**, Xintao Wang**[6]**, Yujiu Yang**[1]**, Chao Dong**[2,3,]
[1] Shenzhen International Graduate School, Tsinghua University
[2] Shenzhen Institutes of Advanced Technology, Chinese Academy of Sciences
[3] Shanghai AI Laboratory     [4] The University of Sydney
[5] University of Chinese Academy of Sciences     [6] ARC Lab, Tencent PCG
`jinjin.gu@sydney.edu.au, ssw20@mails.tsinghua.edu.cn,`
`{chao.dong, lb.xie}@siat.ac.cn, xintaowang@tencent.com`
`yang.yujiu@sz.tsinghua.edu.cn`

## 1    Patch Alignment for Recurrent-based VSR Transformer

The proposed patch alignment method can also be applied to the recurrent VSR framework. Recurrent VSR methods [2, 3, 13] use bidirectional propagation scheme to maximize information gathering in VSR and have achieved the state-of-the-art performance. By replacing the CNN backbone with the Transformer backbone, we can easily build a recurrent VSR Transformer. We employ the second-order grid propagation framework similar to BasciVSR++ [3], where the intermediate features are propagated both forward and backward in an alternating fashion. Through propagation, information from different frames can be used for feature refinement. We replace the feature propagation bock with the MFSAB blocks presented in the main text. The architecture of this recurrent VSR Transformer is shown in Figure 2.

Alignment modules are not absent in the existing recurrent methods. In each feature propagation block, features from different frames are aligned to extract information from the adjacent frames, improving feature expressiveness. BasicVSR [2] uses flow-based alignment method for both images and features and BasicVSR++ [3] uses flow-guided deformable convolution (FGDC) alignment. The proposed patch alignment is also compatible with this architecture. We test different alignment methods on the recurrent VSR Transformer; the results are shown in Table 1. In the recurrent VSR Transformers in this Table, we use 12 MFSABs with shortcut connections every 3 MFSABs for each feature propagation block. The feature size is set to 100, and the number of attention heads is 4. The baseline is the original BasicVSR++ model that uses FGDC and CNN backbone. Replacing the CNN with Transformer blocks can bring a PSNR improvement of 0.5dB on the REDS test set. However, the FGDC alignment used 7.8M parameters, accounting for almost half of all parameters. Replacing the FGDC alignment with the proposed patch alignment achieves competitive results without introducing additional parameters – our method saves 7.8M of parameters. This experiment illustrates the effectiveness of the proposed patch alignment method.

In the main text, we report that the proposed PSRT-recurrent trained using 16 frames demonstrates the state-of-the-art performance on the VSR task, even compared with BasicVSR++, which was trained using 30 frames. We also tried using 30 frames when training PSRT-recurrent. The training curve is shown in Figure 3. It can be seen that PSRT-recurrent can still be greatly improved from more training frames. However, training with 30 frames takes much longer time than with 16 frames, which makes this method uneconomical. In Table 4 we also compare the state-of-the-art contemporaneous work RVRT [7]. When

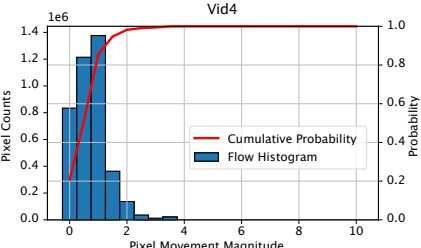

Figure 1: The distribution of the movement for the Vid4 [9] test sets.

36th Conference on Neural Information Processing Systems (NeurIPS 2022).

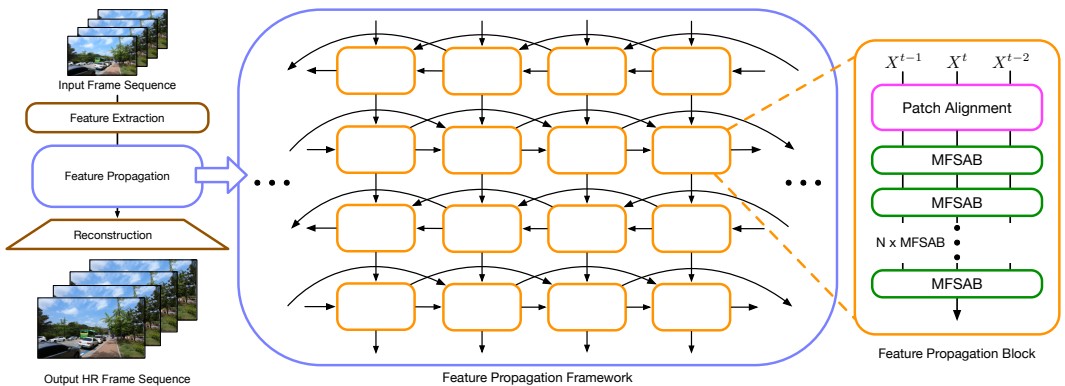

Figure 2: The framework of the used recurrent-based VSR Transformer.

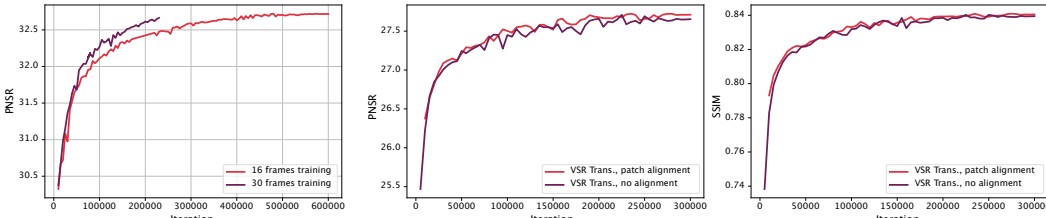

Figure 3: The comparison of with different training frames. Training with 30 frames leads to better performance at the cost of a larger training cost.

Figure 4: We compare the patch alignment method and no alignment on the Vimeo-90K dataset. In this case of small movements, no alignment can already achieve good performance, and patch alignment will not bring much improvement. This shows that Transformers can directly handle a small range of misalignment.

RVRT uses 30 frames for training, it can achieve similar performance to PSRT-recurrent when it was trained with 16 frames. This also demonstrates the superior performance of our method.

## 2 Patch Alignment for the Vimeo-90K Dataset

Unlike the REDS [10] dataset, the motion in the Vimeo-90K dataset [14] tends to be smaller. According to Figure 2 in the main text, movement magnitudes of 98% pixels in the Vimeo-90K dataset are less than 8. As shown from Table 1 in the main text, VSR Transformer without alignment can outperform other alignment methods on this dataset. A natural question is whether the proposed patch alignment is still effective for a small-motion dataset. We conduct an ablation study on the Vimeo-90K and the Vid4 dataset. Table 2 reports the results. We also show the training curve of these two methods in Figure 4 and the distribution of pixel movement for the Vid4 [9] test set in Figure1. As can be observed from the distribution of movement, the Vid4 dataset contains no pixels whose movement magnitudes are larger than 5. Since most of the motion in these two datasets is within the range that VSR Transformer can handle, there is no significant difference between patch alignment and no alignment method, even no alignment version performs slightly better on Vimeo-90K. The validation curve also demonstrates that patch alignment and no alignment show comparable performance. This experiment confirms our conclusions that (1) we can get good results using VSR Transformers without additional alignment for a specific range of misalignment, and (2) the proposed patch alignment does not introduce as many negative effects as other alignment methods.

Finally, we present the results of recurrent VSR Transformer with patch alignment method trained using 14 frames in Table 4. This method achieves state-of-the-art performance on both the Vimeo-90K test set and the Vid4 test set.

## 3 The FLOPs and Runtime of the Proposed Method

We calculate the average FLOPs of our method and some existing methods. This FLOPs is calculated using LR frames with size $180 \times 320$. We also record their average runtime. The results are shown in Table 3. As can be seen, the number of parameters of our method is less than other Transformer methods. One of the reasons is that our method saves a lot of parameters on the alignment module.

Table 1: Ablation study on the different alignment methods and backbone networks. The results are tested on REDS4 [10] dataset for $4\times$ video super-resolution on RGB channels.

| Method | Frames | Params(M) | PSNR | SSIM |
|---|---|---|---|---|
| BasicVSR++ [3], The baseline model | 6 | 7.3 | 31.38 | 0.8898 |
| Flow-guided Deformable Alignment + Transformer | 6 | 18.6 | 31.89 | 0.8967 |
| Patch Alignment + Transformer | 6 | 10.8 | 31.88 | 0.8964 |

Table 2: Ablation study of patch alignment method trained using the Vimeo-90K dataset. The results are tested on Vimeo-90K-T and Vid4 for $4\times$ video super-resolution on Y channel. The experiments with 7 training frames were only trained from scratch for 300K iterations.

| Method | Frames | Params | Vimeo-90K-T | | Vid4 | |
|---|---|---|---|---|---|---|
| | | | PSNR | SSIM | PSNR | SSIM |
| PSRT-recurrent *w/o* alignment | 7 | 12.0 | 37.87 | 0.9508 | 27.71 | 0.8403 |
| PSRT-recurrent | 7 | 13.4 | 37.80 | 0.9502 | 27.72 | 0.8409 |

Our FLOPs and runtime are also within a reasonable range. As the acceleration and optimization of Transformers are still to be studied, we believe that given our relatively small FLOPs, there is room for further optimization of the runtime of our method. For the training time, only VRT reports their training time. VRT need 15 days to train and the proposed PSRT-recurrent needs 18 days to train. Our method's training time and cost are roughly the same compared with VRT.

## 4 Discussion about VSRT

We notice an exception in the effect of the alignment module to a VSR Transformer, *i.e.*, the VSRT model. Although the VSRT model [1] also employs Transformer as the backbone design, the alignment module is necessary for it. Removing alignment in VSRT introduces severe performance degradation. This conflicts with our conclusion. Our discussion on this issue is as follows. The VSRT uses a token size of $8 \times 8$. In the VSRT, self-attention is calculated between different tokens. This calculation is free of the indicative bias of locality. But within the $8 \times 8$ token, only convolution layers and MLP layers participate in the calculation. This calculation is subject to locality bias. If the $8 \times 8$ token is not well-aligned, the convolution layers and MLP layers cannot handle unaligned video frame tokens, and self-attention between tokens cannot help improve this. Therefore, the situation of VSRT does not conflict with the argument of this paper.

## 5 Limitation

Our work discusses alignment in the VSR task, whose downsampling operation leads to unique LR patterns. We believe that other downsampling methods will have similar effects, such as blurring and directly downsampling (the "BD" downsampling in other papers). For these downsampling methods, the conclusions of this paper are still valid. Some of the observations may not apply to other video restoration tasks, because the multi-frame information they need to use may differ. For other video restoration tasks, we believe that the proposed method will still lead to improvement since theoretically patch alignment preserves more information. But if patch alignment is applied without modification, the resulted improvement may not be as big as in the VSR task. Because the nature of the sub-pixel information will change for other applications, the network design can also be changed (such as adding multi-scale designs). We only have limited space in this paper. However, we emphasise the importance of research in this direction and reserve it for future work.

## 6 More Experiments

We show more visual comparisons between the existing VSR methods and the proposed recurrent VSR Transformer with the patch alignment method. We use 16 frames to train on the REDS dataset and seven on the Vimeo-90K dataset. Figure 5 shows the visual results. It can be seen that, in addition to its quantization improvement, the proposed method can generate visually pleasing images with sharp edges and fine details, such as horizontal bar patterns of buildings and numbers on license plates. In contrast, existing methods suffer from texture distortion or loss of detail in these scenes.

Table 3: The comparison of the parameter numbers, FLOPs and the runtime for different methods.

| Method | Parameters (M) | FLOPs (T) | Runtime (ms) |
|---|---|---|---|
| DUF | 5.8 | 2.34 | 974 |
| RBPN | 12.2 | 8.51 | 1507 |
| EDVR [12] | 20.6 | 2.95 | 378 |
| VSRT [1] | 32.6 | 1.6 | – |
| VRT [5] | 35.6 | 1.3 | 243 |
| PSRT-recurrent (Ours) | 13.4 | 1.5 | 812 |

Table 4: Quantitative comparison (PSNR↑ and SSIM↑) on the REDS4 [10] dataset, Vid4 [9], Vimeo-90K-T [14] dataset for $4\times$ VSR task. Red indicates the best and blue indicates the second best performance (best view in color) in each group of experiments.

| Method | Frames REDS/Vimeo | Params (M) | REDS4 | | Vimeo-90K-T | | Vid4 | |
|---|---|---|---|---|---|---|---|---|
| | | | PSNR | SSIM | PSNR | SSIM | PSNR | SSIM |
| Bicubic | -/- | - | 26.14 | 0.7292 | 31.32 | 0.8684 | 23.78 | 0.6347 |
| RCAN [15] | -/- | - | 28.78 | 0.8200 | 35.35 | 0.9251 | 25.46 | 0.7395 |
| SwinIR [6] | -/- | 11.9 | 29.05 | 0.8269 | 35.67 | 0.9287 | 25.68 | 0.7491 |
| TOFlow [14] | 5/7 | - | 27.98 | 0.7990 | 33.08 | 0.9054 | 25.89 | 0.7651 |
| DUF | 7/7 | 5.8 | 28.63 | 0.8251 | - | - | 27.33 | 0.8319 |
| PFNL | 7/7 | 3.0 | 29.63 | 0.8502 | 36.14 | 0.9363 | 26.73 | 0.8029 |
| RBPN | 7/7 | 12.2 | 30.09 | 0.8590 | 37.07 | 0.9435 | 27.12 | 0.8180 |
| EDVR [12] | 5/7 | 20.6 | 31.09 | 0.8800 | 37.61 | 0.9489 | 27.35 | 0.8264 |
| MuCAN [4] | 5/7 | - | 30.88 | 0.8750 | 37.32 | 0.9465 | - | - |
| VSR-T [1] | 5/7 | 32.6 | 31.19 | 0.8815 | 37.71 | 0.9494 | 27.36 | 0.8258 |
| PSRT-sliding | 5/- | 14.8 | 31.32 | 0.8834 | - | - | - | - |
| VRT | 6/- | 30.7 | 31.60 | 0.8888 | - | - | - | - |
| PSRT-recurrent | 6/- | 10.8 | 31.88 | 0.8964 | - | - | - | - |
| BasicVSR [2] | 15/14 | 6.3 | 31.42 | 0.8909 | 37.18 | 0.9450 | 27.24 | 0.8251 |
| IconVSR [2] | 15/14 | 8.7 | 31.67 | 0.8948 | 37.47 | 0.9476 | 27.39 | 0.8279 |
| BasicVSR++ [3] | 30/14 | 7.3 | 32.39 | 0.9069 | 37.79 | 0.9500 | 27.79 | 0.8400 |
| VRT | 16/7 | 35.6 | 32.19 | 0.9006 | 38.20 | 0.9530 | 27.93 | 0.8425 |
| RVRT [7] | 30/14 | 10.8 | 32.75 | 0.9113 | 38.15 | 0.9527 | 27.99 | 0.8462 |
| PSRT-recurrent | 16/14 | 13.4 | 32.72 | 0.9106 | 38.27 | 0.9536 | 28.07 | 0.8485 |

# 7 Detail of Experiments

We present details of the experiments involved in this paper so that anyone can reproduce our results.

**Figure 2 and Figure 1** illustrate the distribution of the movement for three datasets used in our work: Vimeo-90K test set [14], REDS [10] test set and Vid4 [9] test set. We use the pre-trained SypNet [11] to calculate the optical flow. For clips in the Vimeo-90K and Vid4 test set, we measure the motion of the 4th and the 7th frames. For clips in the REDS test set, we measure the motion of the 3rd and the 5th frames. This arrangement is related to the common usage of sliding-window-based VSR models on these datasets: we use seven frames as input on Vimeo-90K, while we only use five frames on REDS. The optical flow result contains two maps, which are the movement in the x-direction $\mathcal{W}^x \in \mathbb{R}^{H \times W}$ and the movement in the y-direction $\mathcal{W}^y \in \mathbb{R}^{H \times W}$. We use the magnitude to indicate the movement of each pixel

$$\mathcal{W}^m_{i,j} = \sqrt{|\mathcal{W}^x_{i,j}|^2 + |\mathcal{W}^y_{i,j}|^2}, \quad \mathcal{W}^m \in \mathbb{R}^{H \times W}.$$

**Figure 3** shows the variation curve of the total variation of the fine-tuned optical flow during training. The total variation is often used to indicate how smooth the optical flow is. The total variation of noise-contaminated optical flow is significantly larger than that of the noise-free optical flow. Given the optical flow $\{\mathcal{W}^x, \mathcal{W}^y\}$, the total variation is calculated as

$$\text{total variation} = \frac{1}{2HW} \sum_{i=1}^{H} \sum_{j=1}^{W} (|\mathcal{W}^x_{i,j-1} - \mathcal{W}^x_{i,j}| + |\mathcal{W}^y_{i+1,j} - \mathcal{W}^y_{i,j}|).$$

We calculate the total variation every 5000 iterations.

**Figure 4** shows the performance differences between VSR Transformers with and without alignment modules for different pixel movements. The VSR Transformer backbone used in this figure contains 16 Multi-Frame Self-Attention Blocks (MFSABs). Similar to SwinIR [6], we add shortcut connections every 4 MFSABs. The feature dimension is 120, and the head number of the multi-head self-attention is 6. To plot the differences, we first partition the pixels into different groups according to their movement conditions and then calculate the mean square error for each group. We subtract the mean square errors of the VSR Transformer with alignment from the errors of the VSR Transformer without alignment. Thus, the parts greater than zero indicate better performance without alignment.

For the first sub-figure, we study the image alignment. The window size is set to 8. We keep the other settings for the second sub-figure and enlarge the window size to 12. For the third sub-figure, we replace the image alignment to feature alignment. In addition to the 2D convolution feature extraction, we add one CNN residual block to extract deep features. These experiments are performed under the same training settings described in Section 3 in the main text.

**Table 1** shows the quantitative comparison of different VSR methods. For VSR CNNs, we use ten residual blocks [8] to extract features for all the input frames. We concatenate the features and reduce the channel number using a convolution layer. Five residual blocks are then used to conduct further processing. The VSR Transformers involved in this table share similar backbone architecture with the VSR Transformers in Figure 4, which contains 16 MFSABs with shortcut connections every 4 MFSABs. The feature dimension is 120, and the head number of the multi-head self-attention is 6. The training method is the same as described in section 3 of the main text. For the methods in which the flow network is not fixed, the learning rate for the flow network is $2.5 \times 10^{-5}$. For the first 5,000 iterations, the flow network is fixed.

**Figure 5** shows the curves of some of the methods in Table 1. One can refer to Table 1 for the experimental details. The calculation of the movement distribution is the same as in Figure 2. The only difference is that we show the percentage in Figure 5, not the counts of pixels.

**Table 2 and Table 3** share the same training setting and Transformer backbone. The VSR Transformer backbone contains 36 MFSABs with shortcut connections every 6 MFSABs. The window size is set to 8. The channel size of the transformer and head size are set to 144 and 6. The training method is the same as described in Section 3 of the main text. For the methods in which the flow network is not fixed, the learning rate for the flow net is $5 \times 10^{-5}$. For the first 20,000 iterations, the flow network is fixed. The implementation of different alignment methods is the same as described in Section 3 of the main text.

**Table 4** shows the quantitative comparison between the proposed method and the state-of-the-art VSR methods. Due to the space limit, we only show limited results in the main text; the full version is shown in Table 4.

The architecture of the PSRT-recurrent is shown in Figure 2 and described in Section 1. For each feature propagation block in PSRT-recurrent, we use 18 MFSABs with shortcut connections every 6 MFSABs. The feature size is set to 120, and the number of attention heads is 6.

The PSRT-sliding backbone contains 36 MFSABs with shortcut connections every 6 MFSABs. The window size is set to 8. The channel size of the transformer and head size are set to 144 and 6.

For the PSRT-recurrent with 16 input frames and the PSRT-sliding method, the total training iteration is 600K. The initial learning rate for these experiments is set to $2 \times 10^{-4}$. All other settings remain unchanged. For the PSRT-recurrent trained on Vimeo-90K, we follow [3] to initialize the model with the well-trained model using REDS. We fine-tune it for the other 300K iterations. The initial learning rate is $1 \times 10^{-4}$.

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

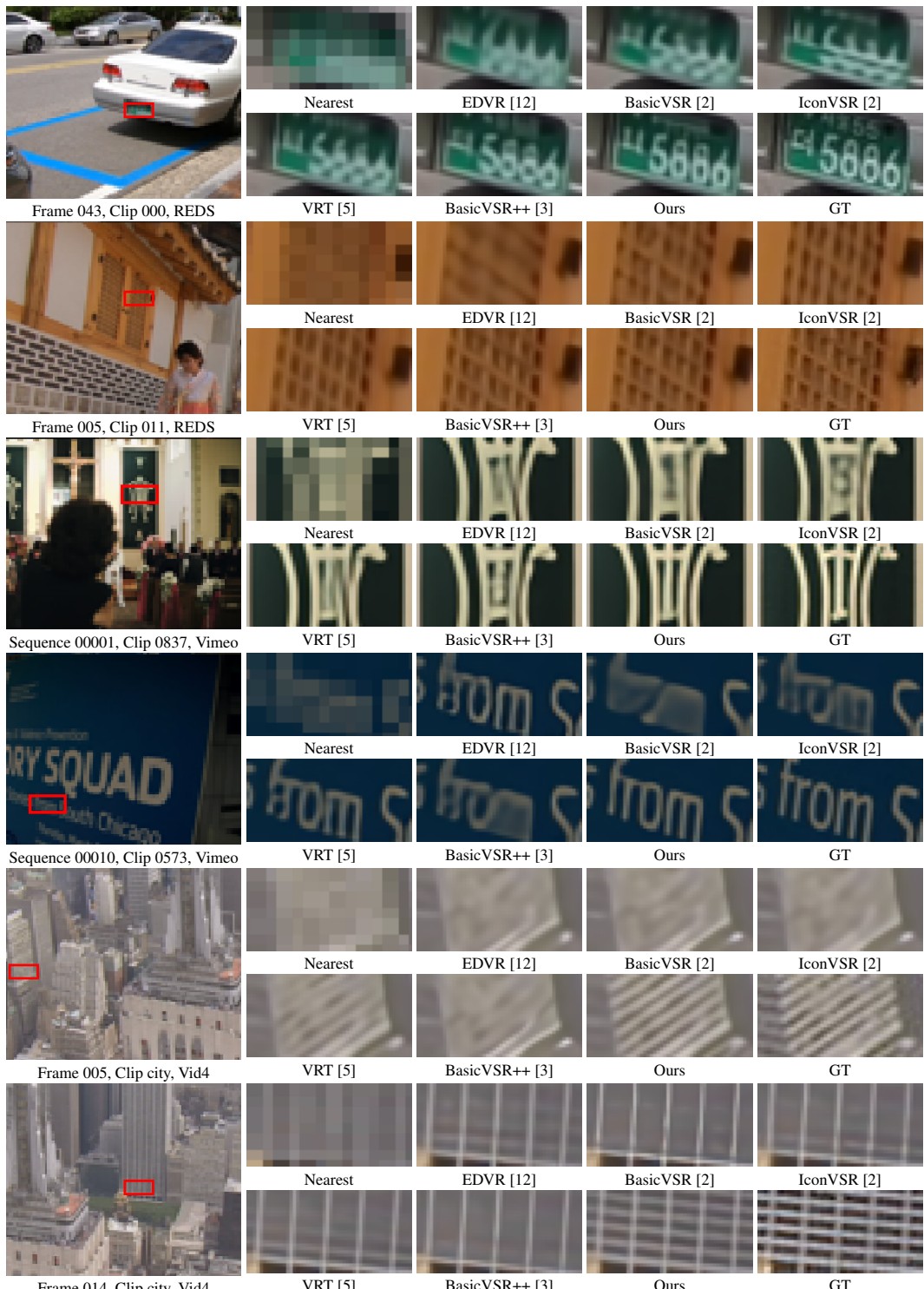

Figure 5: Visual comparison of VSR (×4) on REDS, Vimeo-90K and Vid4 datasets.