# OpenReview forum: "Rethinking Alignment in Video Super-Resolution Transformers"
_NeurIPS.cc/2022/Conference — NeurIPS 2022 Accept_

### Official Review · Reviewer_TWT3 · 2022-07-07

**Rating:** 8
**Confidence:** 4
**Ethics Flag:** Yes
**Soundness:** 3 good
**Presentation:** 3 good
**Contribution:** 3 good

**Summary:**

This paper rethinks alignment in video Super-Resolution Transformers, demonstrates that existing alignment methods are unnecessary or even harmful to the VSR Transformer, and will cause damage to the sub-pixel information, and proposes a simple but effective alignment method.

**Questions:**

1.	It’s not clear about the explanation of Figure 4,how to get the result of Figure 4,especially Figure 4(c).
2.	There should place one line space before the figure caption and one line space after the figure, the layout of the figure is too crowded.

**Limitations:**

This paper use bicubic interpolation to produce LR video frames. It could be further discussed whether the experimental results will be affected by different degradation processes.

**Strengths And Weaknesses:**

Strengths
1.	There is a certain novelty to this work, reconsidering the alignment in video super-resolution Transformers, while other methods retain the complex alignment module
2.	The paper makes a reasonable analysis of the experimental results, proposes that the existing alignment methods are unnecessary or even harmful to the VSR Transformer, and will cause damage to the sub-pixel information.
3.	Proposed a effective alignment method --- patch alignment, and experimental results show that this method achieves state-of-the-art VSR performance.

Weaknesses
1. Details about patch alignment are unclear. Section 5.1 could have further additions, in particular how to find the corresponding patches in the supporting frames for each patch, and how to move the patch to the corresponding position.

---

> ### Author Response · Authors · 2022-08-02
> **Author Responses to Reviewer #4 (TWT3)**
>
> We thank Reviewer #4 for his/her comments and the appreciation of our novelty, experiments and new method. To the concerns expressed by Reviewer #4 in `Weaknesses`,  `Questions` and `Limitations`, our response is as follows:
>
> `R4-Weakness`: Thanks for the suggestion. The proposed method is intuitive, and the method is easy to understand. We are concerned that introducing too many mathematical expressions will hinder the understanding of the method. We use the averaged patch motion vector to locate the corresponding patch in the supporting frame. The calculated motion vector represents the amount of movement of this patch in the supporting frame. We round this vector to avoid possible interpolation operations. Then we cut and move the entire patch found according to this vector to the same position as the patch in the reference frame. Reviewer #4 is encouraged to review Figure 7 and its caption for an intuitive understanding. We will also further improve this part of the presentation.
>
> `R4-Q1`: Thanks for the question. We present more details of the experiments involved in this paper in Section 2 of the Supplementary Material so that anyone can reproduce our results. An illustration of Figure 4 can also be found there. We paste the relevant presentation here for the reviewer to check.
>
> Figure 4 shows the performance differences between VSR Transformers with and without alignment modules for different pixel movements. The VSR Transformer backbone used in this figure contains 16 Multi-Frame Self-Attention Blocks (MFSABs). Similar to SwinIR, we add shortcut connections every 4 MFSABs. The feature dimension is 120, and the head number of the multi-head self-attention is 6. To plot the differences, we first partition the pixels into different groups according to their movement conditions and then calculate the mean square error for each group. We subtract the mean square errors of the VSR Transformer with alignment from the errors of the VSR Transformer without alignment. Thus, the parts greater than zero indicate better performance without alignment. For the first sub-figure, we study the image alignment. The window size is set to 8. We keep the other settings for the second sub-figure and enlarge the window size to 12. For the third sub-figure, we replace the image alignment to feature alignment. In addition to the 2D convolution feature extraction, we add one CNN residual block to extract deep features. These experiments are performed under the same training settings described in Section 3 in the main text.
>
> `R4-Q2`: Thanks for the suggestion. We will increase the space before and after the figures to promote the aesthetics of the paper.
>
> `R4-Limitation`: Thanks for the comment. We also believe that research on different degradations is necessary. The topic of this paper is Video SR, whose downsampling operation leads to unique LR patterns (shown in Figure 10). We believe that other downsampling methods will have similar effects, such as blurring and directly downsampling (the "BD" downsampling in other papers). For these downsampling methods, the conclusions of this paper are still valid. For other video restoration tasks, we believe that the proposed method will still lead to improvement since theoretically patch alignment preserves more information. But if patch alignment is applied directly, the resulted improvement may not be as big as in Video SR. Because the nature of its sub-pixel information will change for other applications, its network design can also be changed (such as adding multi-scale designs). We only have limited space in this paper. We emphasise the importance of research in this direction and reserve it for future work.

---

> ### Author Response · Authors · 2022-08-05
> **Further discussion with Reviewer TWT3 (denoted as R4)**
>
> Dear reviewer TWT3:
>
> We thank you for the precious review time and valuable comments. We have provided corresponding responses and results, which we believe have covered your concerns. We hope to further discuss with you whether or not your concerns have been addressed. Please let us know if you still have any unclear parts of our work.
>
> Best,
>
> Paper 1445 Authors.

---

### Official Review · Reviewer_F7UK · 2022-07-11

**Rating:** 6
**Confidence:** 3
**Soundness:** 3 good
**Presentation:** 4 excellent
**Contribution:** 3 good

**Summary:**

This work explores the alignment modules in Transformer-based video super-resolution (VSR) methods. The research begins with experiments finding that VSR Transformers can directly utilize the unaligned videos and existing alignment methods can harm VSR Transformer. To solve this problem, this paper proposes a new and efficient alignment method named patch alignment, which aligns image patches instead of pixels. The experiments show the proposed method achieves state-of-the-art performance.

**Questions:**

My main concern of this paper is the novelty and contribution. The problem of alignment does not seem broad and relevant, and I'm not sure if this research will contribute enough to the field. Another concern is that I did not find a clear experiment of how much improvement the patch alignment method brings compared to the baseline method, so I’m not sure if it's efficient enough.

**Limitations:**

Yes

**Strengths And Weaknesses:**

Strengths:
1) The paper is easy to follow. The motivation is clear and straight, and the main problem is well defined.
2) The problem of alignment for VSR Transformers is interesting, this paper shows VSR Transformers can utilize the unaligned videos.
3) The performance is state-of-the-art, the proposed method achieves the first or second best performance on each benchmark.
4) The method is efficient, there are not many extra parameters and computations introduced.

Weakness:
1) The main weakness seems to be the limited novelty and contribution. The main contribution of this work is the patch alignment method, but the method of aligning the patch instead of pixel seems too simple. The problems and solutions presented are relatively narrow, and I'm afraid this doesn't contribute much. The problem and solution proposed are relatively narrow, and this may not contribute much.
2) The experiment is not very clear, I see that the VSR Transformers with patch alignment method achieves good performance, but I cannot find how much improvement the patch alignment brings compared with the baseline method.

---

> ### Author Response · Authors · 2022-08-02
> **Author Responses to Reviewer #3 (F7UK) (Part 2/2)**
>
> `R3-Weakness 2` and `R3-Q2`:  Thanks for the question. We have presented the comparison of our methods with baselines. There are two sets of comparisons. One is for the sliding window-based VSR Transformers, and the results are included in Table 2 and Table 3. Another is for the recurrent-based methods, and the results are included in Table 1 of the Supplementary Material. We paste these results here for a quick reference.
>
> **Sliding-window-based methods**:
>
> For the VSR Transformers based on a sliding-window strategy, the proposed patch alignment outperforms state-of-the-art methods based on flow-guided deformable convolution (FGDC) while saving a large number of parameters.
>
> | Exp. Index | Alignment Method              | Alignment Position | Resampling Method | Params (M) | REDS (PSNR/SSIM) |
> |------------|-------------------------------|--------------------|-------------------|------------|------------------|
> | 1          | No                            | --                 | --                | 12.9       | 30.92/0.8759     |
> | 2          | Image alignment               | Image              | Bilinear          | 12.9       | 30.84/0.8752     |
> | 3          | Feature alignment             | Feature            | Bilinear          | 14.8       | 31.06/0.8792     |
> | 4          | Deformable Convolution (FGDC) | Feature            | --                | 16.1       | 31.11/0.8804     |
> | 5          | Patch alignment               | Image              | Nearest Neighbor  | 12.9       | 31.11/0.8800     |
> | 6          | Patch alignment               | Feature            | Nearest Neighbor  | 14.8       | 31.17/0.8810     |
>
> **Recurrent-based methods**:
>
> For the recurrent-based VSR Transformers, the baseline model is the BasicVSR++. As can be seen, replacing the CNN with Transformer blocks can bring a PSNR improvement of 0.5dB on the REDS test set. However, the FGDC alignment used 7.8M parameters, accounting for almost half of all parameters. Replacing the FGDC alignment with the proposed patch alignment achieves competitive results without introducing additional parameters – our method saves 7.8M of parameters. This experiment illustrates the effectiveness of the proposed patch alignment method.
>
> | Method                                         | Frames | Params (M) | REDS PSNR | REDS SSIM |
> |------------------------------------------------|--------|------------|-----------|-----------|
> | BasicVSR++ [3], The baseline model             | 6      | 7.3        | 31.38     | 0.8898    |
> | Flow-guided Deformable Alignment + Transformer | 6      | 18.6       | 31.89     | 0.8967    |
> | Patch Alignment + Transformer                  | 6      | 10.8       | 31.88     | 0.8964    |

---

> ### Author Response · Authors · 2022-08-02
> **Author Responses to Reviewer #3 (F7UK) (Part 1/2)**
>
> We thank Reviewer #3 for his/her comments and the appreciation of our work. To the concerns expressed by Reviewer #3 in `Weaknesses` and `Questions`, our response is as follows:
>
> `R3-Weakness 1` and `R3-Q1`: Thanks for the comment. We respectfully disagree with your opinion on our novelty and topic.
>
> **Novelty**:
>
> First of all, the most important contribution of our work is not the method of patch alignment, but the analysis and conclusion with alignment and VSR Transformer. We use a number of novel analysis methods, such as (1) a case-by-case comparison of the effect of alignment on content under different motion conditions, (2) a novel video super-resolution attribution analysis, (3) the analysis of the distribution shift of fine-tuned optical flow, (4) the analysis of the smoothness of fine-tuned optical flow, and (5) quantitative comparison of the resampling methods in alignment. The finds are also novel to this community. Our findings challenge our common understanding of using Transformers to process multiple spatially misaligned images. The analysis of alignment can provide useful insights for VSR. We need to utilize inter-frame sub-pixel information, yet many image pre-process operations interfere with our utilization of this information. These observations hint that Transformer can implicitly make accurate connections for misaligned pixels. Many low-level vision tasks can take advantage of this property, such as video restoration, reference-based SR, burst image processing, stereo matching, flow estimation, etc. When designing Transformers for these tasks, we can no longer explicitly introduce the alignment modules or the cost volume modules, but give full play to the powerful modeling capabilities of the Transformer itself. These statements can be found in L335--L344 of the main text.
>
> Second, the patch alignment is also effective and novel. It is the first time an alignment method is proposed to treat supporting frames as patches and keep pixels within the patch unchanged during processing. Contrary to reviewer #3, we believe it is important to keep the method simple, as this can demonstrate in the clearest way the core reasons for the success of our method. We would like to emphasize that the simple operation of the proposed patch alignment improves its performance on REDS by 0.33dB while saving nearly 2/3 of the parameters compared to the current state-of-the-art VSR Transformer model. This is enough to demonstrate the superiority and importance of the proposed method.
>
> **Note that none of the other three reviewers questioned our novelty**. Especially, Reviewer TWT3 commented: "There is a certain novelty to this work".
>
> **Significance**:
>
> Our response to Reviewer 3's concerns about the subject of this paper is as follows. First, there is no doubt that Video SR is a very important low-level vision task. The core issue of this task is to process sub-pixel information between multiple frames. In this regard, alignment and information transfer are the most important research topics. A large number of related works have been proposed to discuss the most efficient alignment method [4, 5, 6, 12, 18, 20, 21, 24, 35, 36, 40, 41, 44, 48, 49] for the VSR task. In state-of-the-art VSR models, the alignment module is also usually the most complex and bulky module. For example, alignment occupies 1/3 of the parameters in VRT, and 1/3 of the parameters in BasicVSR++. Our work presents novel insights in this regard. The proposed method optimizes the overly complex alignment design of the VSR Transformers [3, 21] into a near-zero-cost operation. This has demonstrated the importance of our research.

---

> ### Author Response · Authors · 2022-08-05
> **Further discussion with Reviewer F7UK (denoted as R3)**
>
> Dear reviewer F7UK:
>
> We thank you for the precious review time and valuable comments. We have provided corresponding responses and results, which we believe have covered your concerns. In particular, we would like to address your concerns about our novelty. We hope to further discuss with you whether or not your concerns have been addressed. Please let us know if you still have any unclear parts of our work.
>
> Best,
>
> Paper 1445 Authors.

---

### Official Review · Reviewer_j86q · 2022-07-11

**Rating:** 7
**Confidence:** 4
**Soundness:** 2 fair
**Presentation:** 3 good
**Contribution:** 2 fair

**Summary:**

This paper rethinking the alignment in video super-resolution (VSR) Transformer and has two observations. (i) VSR Transformers can directly utilize multi-frame information from unaligned videos, and (ii) existing alignment methods are sometimes harmful to VSR Transformers. Based on these two observations, the authors remove the alignment in VSR, and propose a patch alignment to reduce the computational burden. Many experiments demonstrate the state-of-the-art performance on REDS.

**Questions:**

1. The authors state that "existing alignment methods are sometimes harmful to VSR Transformers". The importance of the alignment has been proved in most existing VSR methods. The performance of the alignment highly depends on the specific architecture of VSR Transformers. The observation may be only suitable for the proposed Transformer in this paper.

2. The proposed VSR Transformer implicitly conduct the alignment and track the motion between frames. The authors provide visualization of local attribution map to highlight the pixels in each frame. It would better to explain from mechanism why the proposed VSR Transformer implicitly can align the feature in each frame.

3. Recently, some studies demonstrate that large window size has better performance. In Table 4, the performance gain on REDS4 may mainly come from the large window size of 16x16. To verify the effectiveness of the patch alignment. It would better to report results with the window size of 8 in the supplementary.

4. Patch alignment is sensitive to the similarity metric. When the adjacent frames has different illumination and scale, the performance of patch alignment may drop. How to relieve this issue in the experiment.

5. The authors state that the proposed method can del with large motions. However, the performance is worse than VRT on Vimeo-90K-T which has different degrees of motions. Vimeo training set has 7 frames

6. How many days for training PSRT-recurrent on the RED dataset? Besides, it would be better to report the inference time.

**Limitations:**

Yes

**Strengths And Weaknesses:**

Strengths:
1. The authors conduct extensive experiments to explore the role of the alignment in VSR and have two observations.
2. The proposed VSR Transformer method achieves the state-of-the-art performance on multiple datasets.

Weaknesses:
1. For the first observation, VSR-T and VRT are able to  utilize multi-frame information.
2. The second observation "existing alignment methods are sometimes harmful to VSR Transformers" is two strong.

---

> ### Author Response · Authors · 2022-08-02
> **Author Responses to Reviewer #2 (j86q) (Part 3/3)**
>
> `R2-Q5`: Thanks for the question. The test set of Vimeo-90K is divided into three test sets, slow, medium, and fast. But according to our statistics, even the Vimeo-90K fast dataset's motion is far from the REDS test set's motion. The relevant statistical histogram is shown in Figure 2, and its specific measurement method is described in Section 2 of the Supplementary Material. As can be seen that the motion in the Vimeo-90K dataset is generally small and movement magnitudes of 99% pixels are less than 10 (for each clip, we measure the motion of the 4th and the 7th frames). Differently, there are large motions in the REDS dataset. There are at least 20% of pixels that have movement magnitudes larger than 10 (for each clip, we measure the motion of the 3rd and the 5th frames). Therefore, if we want to compare the performance of the patch alignment method under large motion conditions, the performance on the REDS dataset is more convincing. We paste some comparisons here for quick reference. Note that our largest model PSRT-recurrent was not completely trained at the time of submission. The performance we paste here is the final performance of our model. RVRT [R2] is the state-of-the-art VSR model released days before. As can be seen that we achieve state-of-the-art performance on the REDS dataset. When trained using 7 frames, we achieve the best performance on the Vid4 test set and comparable performance on the Vimeo-90K test set. When trained using 14 frames, we achieve the best performances on both Vid4 and Vimeo-90K test sets. The experimental results demonstrate the superior performance of the proposed patch alignment method. We will update these results in the revised paper.
>
> | Method         | Frames (REDS/Vimeo) | Params (M) | REDS (PSNR/SSIM) | Vimeo-90K (PSNR/SSIM) | Vid4 (PSNR/SSIM) |
> |----------------|---------------------|------------|------------------|-----------------------|------------------|
> | BasicVSR++     | 30/14               | 7.3        | 32.39/0.9069     | 37.79/0.9500          | 27.93/0.8425     |
> | VRT            | 16/7                | 35.6       | 32.19/0.9006     | 38.20/0.9530          | 27.93/0.8425     |
> | PSRT-recurrent | 16/7                | 13.4       | 32.72/0.9106     | 38.15/0.9529          | 28.03/0.8496     |
> | RVRT [R2]      | -/14                | 10.8       | -                | 38.15/0.9527          | 27.99/0.8462     |
> | PSRT-recurrent | -/14                | 13.4       | -                | 38.27/0.9536          | 28.07/0.8485     |
>
> `R2-Q6`: Thanks for the question. We paste some of the data here for a quick reference. We calculate the average FLOPs of our method and some existing methods. This FLOPs is calculated using LR frames with size $180\times320$. We also record their average runtime of them. As can be seen, the number of parameters of our method is less than other Transformer methods. One of the reasons is that our method saves a lot of parameters on the alignment module. Our FLOPs and runtime are also within a reasonable range. As the acceleration and optimization of Transformers are still to be studied, we believe that given our relatively small FLOPs, there is room for further optimization of the runtime of our method. For the training time, only VRT reports their training time. Our method's training time and cost are roughly the same compared with VRT. We will include these results and a discussion of FLOPs and runtime in the revised paper.
>
> | Method         | Training Days | Parameters (M) | FLOPs (T) | Runtime (ms) |
> |----------------|---------------|----------------|-----------|--------------|
> | DUF            | -             | 5.8            | 2.34      | 974          |
> | RBPN           | -             | 12.2           | 8.51      | 1507         |
> | EDVR           | -             | 20.6           | 2.95      | 378          |
> | VSRT           | -             | 32.6           | 1.6       | -            |
> | VRT            | 15            | 35.6           | 1.3       | 243          |
> | PSRT-recurrent | 18            | 13.4           | 1.5       | 812          |

---

> ### Author Response · Authors · 2022-08-02
> **Author Responses to Reviewer #2 (j86q) (Part 2/3)**
>
> `R2-Q1`: Thanks for your question. First of all, the VSR Transformer architecture used in this research is based on a general image processing Transformer structure, and its effectiveness has been proven in many recent research works [21, 22, 45, 49, R1, R2]. This kind of Transformer structure contains a single-pixel token and window-based attention combined with a shift-window mechanism. In addition, all Transformer configurations used in our research are the most basic and extensive, thus ensuring our conclusions' generality.
>
> In response to the reviewer's concerns, we did find that a structure similar to VSRT may not be within the scope of our discussion. The reason is already explained in `R2-Weakness 1`. If a large amount of spatial information (e.g., $8\times8$ tokens) needs to be processed inside the token, self-attention cannot help extract multi-frame information. But this class of designs, e.g., VSRT and IPT, have proven to be inefficient for image processing tasks [21, 22, R1, R2]. Thus this issue does not affect the importance of our conclusions. We sincerely thank the reviewer for bringing this question to us, and we will add this discussion to the revised paper.
>
> Additional References:
>
> [R1] Chen, X., Wang, X., Zhou, J. and Dong, C., 2022. Activating More Pixels in Image Super-Resolution Transformer. arXiv preprint arXiv:2205.04437.
>
> [R2] Liang, J., Fan, Y., Xiang, X., Ranjan, R., Ilg, E., Green, S., Cao, J., Zhang, K., Timofte, R. and Van Gool, L., 2022. Recurrent Video Restoration Transformer with Guided Deformable Attention. arXiv preprint arXiv:2206.02146.
>
> `R2-Q2`: Thanks for the suggestion. Our local attribution map visualization results indicate that VSR Transformer implicitly estimates the motion between frames and inferences based on the moving objects. We believe that this capability of the Transformer is related to the computation of self-attention. In self-attention layers, we first calculate the attention matrix between the pixel tokens within the attention window. Pixels representing the same object may have higher attention scores. These scores guide the fusion between the features from the reference frame and the supporting frames. Thus, only related pixels are selectively calculated, which plays a similar role to alignment. We include a relative description in L250 of the main text. We will refine this part of the explanation in the revised paper.
>
> `R2-Q3`: Thanks for the question. We listed all the experimental details in Section 2 of the Supplementary Material. The experiments in Table 4 are implemented with a window size of 8 and can already achieve state-of-the-art performance. In fact, only one VSR Transformer experiment in Table1 uses a window size of 16. For the rest of the quantitative experiments, we all use a window size of 8. We will make this clearer in the revised paper.
>
> `R2-Q4`: Thanks for the question. In patch alignment, we use the estimated optical flow to find the best matching patch, rather than a similarity metric. For the problem of "different illumination and scales" mentioned by reviewer #2, the other alignment methods based on motion estimation will also encounter challenges, not just patch alignment. Also, since patch alignment uses the averaged motion vector within the patch to find the corresponding one, the averaging operation can introduce additional robustness. So patch alignment is affected by this situation less than other methods. The excellent performance of our method also speaks to the same conclusion.

---

> ### Author Response · Authors · 2022-08-02
> **Author Responses to Reviewer #2 (j86q) (Part 1/3)**
>
> We thank Reviewer #2 for his/her comments and the appreciation of our work. To the concerns expressed by Reviewer #2 in `Weaknesses` and `Questions`, our response is as follows:
>
> `R2-Weakness 1`:  Thanks for your comment. This paper argues that "VSR Transformers can **DIRECTLY** utilize multi-frame information from **UNALIGNED** videos". Here, we highlight two important conditions in this argument. VRT and VSRT both use a complex alignment design to align content in different frames. Although they can also utilize information from multiple frames to a certain extent, this exploitation is not DIRECTLY exploited from UNALIGNED video frames.
>
> For the VRT model, we want to emphasise that its alignment module takes 1/3 of all the parameters. With this alignment module, the VRT is not taking advantage of the Transformer's ability to handle misaligned frames. The loss of sub-pixel information caused by this alignment module may further decrease the performance of the VRT. In other words, the design of the VRT does not benefit from the insight described in this paper.
>
> For the VSRT model, it uses a token size of $8\times8$. In the VSRT, self-attention is calculated between different tokens. But within the $8\times8$ token, only CNN and MLP participate in the calculation. For VSRT, the function of utilizing multi-frame information is mostly done by CNN and MLP within the token, rather than self-attention computed between tokens. If the $8\times8$ token is not well-aligned, the CNN and MLP cannot handle unaligned video frame tokens, and self-attention between tokens cannot help improve this.
>
> Therefore, the situations of VRT and VSRT do not conflict with the argument of this paper. On the contrary, both two cases demonstrate the value of our work. The knowledge and insights presented in this paper can help us understand the problems that still exist in VRT and VSRT design. It can also inspire future VSR models, such as the patch alignment method proposed in this paper.
>
> `R2-Weakness 2`: Thanks for the comment. We agree that making absolute assertions such as "all alignment is harmful" may be too strong, since we can't try all existing and future alignment methods. But our claim is that "EXISTING alignment methods are SOMETIMES harmful to VSR Transformers". To support our claim, we select existing alignment methods that are recognized for their generality and performance, i.e., image alignment which is a classic and intuitive alignment method and is used in many VSR models; feature alignment which is also well-studied and is the basis of many other alignment methods; and deformable convolution based alignment that supports the state-of-the-art VSR models such as EDVR, BasicVSR++. Our conclusions are validated by these representative EXISTING alignment methods. We found that in the range of motion that the VSR Transformers can already handle well, using alignment reduces the VSR performance it could otherwise achieve. In the case of large movements, we do not deny the role of alignment. In conclusion, we believe that our summary of the phenomena we have found is appropriate and not too strong.

---

> ### Author Response · Authors · 2022-08-05
> **Further discussion with Reviewer j86q (denoted as R2)**
>
> Dear reviewer j86q:
>
> We thank you for the precious review time and valuable comments. We have provided corresponding responses and results, which we believe have covered your concerns. We hope to further discuss with you whether or not your concerns have been addressed. Please let us know if you still have any unclear parts of our work.
>
> Best,
>
> Paper 1445 Authors.

---

> > ### Comment · Reviewer_j86q · 2022-08-05
> > **Update rating**
> >
> > Thank you for your responses. The authors address my concern. I am happy to increase my rating.

---

### Official Review · Reviewer_CC5y · 2022-07-11

**Rating:** 6
**Confidence:** 5
**Soundness:** 3 good
**Presentation:** 3 good
**Contribution:** 3 good

**Summary:**

This paper studies the alignment problem in video super-resolution. It conducts a series of experiments to show that attention plays a key role for the problem. Using larger window size for unaligned videos can also achieve good performance. Besides, to reduce computation cost, it proposes a patch alignment strategy that aligns different patches. It achieves competitive performance on multiple video super-resolution benchmark datasets.


**Questions:**

See weaknesses.

**Ethics Review Area:**

["I don’t know"]

**Limitations:**

The authors have discussed the limitations and potential societal impact.


**Strengths And Weaknesses:**

Strengths:

1, interesting finding on the relationship between attention window size and motion magnitude.

2, a patch alignment strategy that crops and aligns patches rather than pixels





Weakness:

1, Previous attention-based method [21] proposes to use attention for implicit alignment, along with which feature alignment is used to deal with misalignment outside of the attention window. The key idea seems similar: attention can deal with misalignments.

2, When the motion is small, the feature after alignment is largely still within the attention window. Why no alignment is better for small motions? Is there any more evidence (except for the final PSNR comparison)? Besides, due to the shifted window strategy, one pixel should have direct interactions with its neighbouring 12x12 pixels, why no alignment is worse when pixel movement magnitude is larger than 8 in Fig. 4 (a)?

3, If alignment is unnecessary, global attention (largest window size) should achieve best performance.  Why does the PSNR of [3] (using global attention) is far from good? How about the performance of using global attention for the proposed framework?

4, The patch alignment strategy is very similar to [24], which also crops patches from image features. [24] is even more flexible than this paper, as it samples several patches at a time. Attention is used in a similar way.

5, The patch alignment strategy is also very similar to [1] that samples tokens (patches) for attention according to optical flow.

6, Contradictory experiments results. When using nearest sampling in warping, the optical flow cannot be updated during training as there is no gradient. According to your conclusion, using an updated smooth flow is better than using the original pre-trained flow.

7, If we need to find corresponding patches for each window, do we need to do it again when the windows are shifted?

8, Some conclusions are not new. For example, the superiority of feature warping to image warping is already well-studied in previous works.

9, No runtime and FLOPS comparison.

---

> ### Author Response · Authors · 2022-08-01
> **Author Responses to Reviewer #1 (CC5y) (Part 3/3)**
>
> `R1-Q5`:  Thanks for the question. The [1] citation of the main text is the Layer Norm paper, and the citation [1] of the supplementary material is VSRT. Here we will answer the reviewer's question based on VSRT. In `R1-Q3`, we discussed the differences between the VSR Transformer architecture used in our work and VSRT. The patch in VSRT refers to treating the $8\times8$ patch as a whole and as a token. There is no self-attention operation inside their patch token, only the operation of CNN and MLP. Their self-attention operations are performed between different patch tokens. Our method is to divide the $8\times8$ patch into an attention window. In each attention window, we use each pixel as a token and perform self-attention operations between different pixel tokens. The implicit alignment described in this paper is performed between tokens that participate in self-attention. There is no self-attention inside the token, so alignment is still required for $8\times8$ token in the VSRT.
>
> `R1-Q6`: Thanks for the question. We argue that our results are not contradictory. Warping with NN resampling is equivalent to rounding the optical flow to a certain extent. At this time, no matter whether the optical flow is smooth or not, the difference after rounding will not be too large. And what we describe as "smooth optical flow is better" holds in the case of warping with bilinear resampling. In this case, the smooth optical flow introduces less high-frequency noise. There are also fewer cases where adjacent pixels are warping with very different optical flows. This helps preserve sub-pixel information for high performance.
>
> `R1-Q7`: Thanks for the question. We only do the patch alignment operation once in the proposed method. After the shift operation, discontinuities between patches in the supporting frame will appear within the scope of self-attention, as shown in Figure 8. Indeed, preserving this discontinuity in the supporting frames is not intuitive. But we have found that this simple method already gives a good result. This shows that the information retained by our patch alignment is very important. We also retain the possibility to further improve the performance of patch alignment through more complex designs.
>
> `R1-Q8`: Thanks for your comment. We also believe that the conclusions about image alignment and feature alignment are not new. So we do not claim that this discussion is one of the contributions of this paper. The findings of this paper are consistent with this widely accepted conclusion. Our work provides an understanding of this view. Feature alignment extracts these patterns before sub-pixel information is corrupted by alignment, and produces better performance.
>
> `R1-Q9`: Thanks for your suggestion. We calculate the average FLOPs of our method and some existing methods. The FLOPs results are calculated using LR frames with size $180\times320$. We also record their average runtime. The results are shown below. As can be seen, the number of parameters of our method is less than other Transformer methods. One of the reasons is that our method saves a lot of parameters on the alignment module. Our FLOPs and runtime are also within a reasonable range. As the acceleration and optimization of Transformers are still to be studied, we believe that given our relatively small FLOPs, there is room for further optimization of the runtime of our method. We will include these results and a discussion about FLOPs and runtime in the revised paper.
>
> | Method | Parameters (M) | FLOPs (T) | Runtime (ms) |
> |--------|----------------|-----------|--------------|
> | DUF    | 5.8            | 2.34      | 974          |
> | RBPN   | 12.2           | 8.51      | 1507         |
> | EDVR   | 20.6           | 2.95      | 378          |
> | VSRT   | 32.6           | 1.6       | -            |
> | VRT    | 30.7           | 1.3       | 243          |
> | Ours   | 13.4           | 1.5       | 812          |
>
> `Other Comments`: Finally, we thank Reviewer #1 for his/her insightful comments and the reviewer's efforts in reviewing this paper.

---

> ### Author Response · Authors · 2022-08-01
> **Author Responses to Reviewer #1 (CC5y) (Part 2/3)**
>
> `R1-Q3`: Thanks for the question. Although the VSRT model presented in [3] has global attention, two designs limit its performance. First, the VSRT uses image warping alignment. According to our conclusion, alignment can corrupt sub-pixel information carried in video frames, resulting in a performance drop. Second, the VSRT uses a token size of $8\times8$. In the VSRT, self-attention is calculated between different tokens. This calculation is free of the indicative bias of locality. But within the $8\times8$ token, only CNN and MLP participate in the calculation. This calculation is subject to locality bias. If the $8\times8$ token is not well-aligned, the CNN and MLP cannot handle unaligned video frame tokens, and self-attention between tokens cannot help improve this.
>
> The situation mentioned by the reviewer is more relevant to using a larger window in our framework. We also present these experiments in Table 1. We copy some of the results here for quick reference. We can see that the improvement of the VSR Transformer with a larger window is limited on the Vimeo-90K benchmark, because the window size of 8 can already handle the motion of Vimeo-90K according to Figure 2. On the contrary, using a larger window can bring significant improvement for the REDS dataset because it can handle a wider range of motion. It is reasonable to speculate that if a VSR Transformer with a global window is built according to the method described in this paper, its performance on REDS will be further improved. But this attempt is uneconomical in engineering, especially when the proposed patch alignment can already solve this problem to some extent.
>
> We thank the reviewer again for bringing this question to us. We will add a discussion about the situation of the VSRT model in the revised version.
>
> | Method          | Alignment | Window Size | Vimeo-90K    | REDS          |
> |-----------------|-----------|-------------|----------------|----------------|
> | VSR Transformer | No        | 8           | 37.43 / 0.9470 | 30.56 / 0.8696 |
> | VSR Transformer | No        | 16          | 37.46 / 0.9474 | 30.81 / 0.8745 |
>
> `R1-Q4`: Thanks for the question. FGST [24] is an excellent concurrent work. We learned a few days before our submission that FGST was accepted by ICML. We congratulate the authors of FGST.
>
> Back to the reviewer's question, although FGST and we use the same words in some expressions, we have some major differences. FGST expresses their method as finding similar patches in the supporting frame, but it is implemented by pixel-wise warping using NN resampling at the deep feature. Due to its multi-scale network design, each pixel in the deep feature corresponds to a patch region in the original image. Aside from the difference in multi-scale design, FGST is somewhat equivalent to NN warping on deep features. However, our proposed method manually divides the patch on the image and keeps the pixels within the patch untransformed during alignment. We emphasize that it is important to maintain invariant relationships between pixels within a patch, which FGST does not demonstrate in this regard. We include a comparison of this approach in Table 2. We copy the results below for quick reference. It can be seen that in the absence of explicit patch partitioning, only using NN warps on deep features such as FGST provides the best performance among existing alignment methods. However, the proposed image level patch alignment can already produce the same performance, and feature level patch alignment performs better than their FGST method.
>
> | Method            | Align. Position | Resampling | REDS (PSNR) | REDS (SSIM) |
> |-------------------|-----------------|------------|-------------|-------------|
> | Feature alignment | Feature         | Bilinear   | 31.06       | 0.8792      |
> | Feature alignment | Feature         | NN         | 31.11       | 0.8801      |
> | Patch alignment   | Image           | NN         | 31.11       | 0.8800      |
> | Patch alignment   | Feature         | NN         | 31.17       | 0.8810      |
>
> On the other hand, our findings can also explain the above experimental results. For the NN warping, after the optical flow is rounded, the values of most of the flat regions (where optical flow does not change drastically) become the same integer. At this point, these regions are no longer subject to the loss of sub-pixel information from inaccurate optical flow and interpolation. Therefore, the performance will be improved compared to other methods. And this operation can cause problems in some areas where the optical flow changes drastically. Patch alignment avoids the problem of the NN warping failure in such areas by dividing patches and keeping the patch pixels unchanged. This not only justifies the proposed patch alignment method, but also illustrates the practical value of our analytical work (Section 4). These findings both explain existing methods and inform new ones.

---

> ### Author Response · Authors · 2022-08-01
> **Author Responses to Reviewer #1 (CC5y) (Part 1/3)**
>
> We thank Reviewer #1 for his/her insightful comments. Our responses are as follows:
>
> `R1-Q1`: Thanks for your question. The VRT paper does not directly demonstrate our argument that "VSR Transformers can directly utilize multi-frame information from unaligned videos". On the contrary, a complex alignment module remains in the design of the VRT, and this alignment module takes 1/3 of all the VRT parameters. With this alignment module, the VRT is not taking advantage of the Transformer's ability to handle misaligned frames. The loss of sub-pixel information caused by this alignment module may further decrease the performance of the VRT. In other words, the design of the VRT does not benefit from the insight described in this paper.
>
> This question raised by reviewer #1 also demonstrates the value of our work. There are three reasons. (1) Contrary to existing explanatory speculation, our work demonstrates the Transformer's superior ability to handle misaligned video frames with well-designed experimental analyses. Previous practices such as VRT and VSRT, although possibly involving similar network structures, were performed on aligned videos. (2) Our work shows through experiments and illustration why existing alignment methods are sometimes harmful to VSR Transformers. This not only guides our development of the patch-alignment method but also guides the design of other VSR methods in the future. (3) Our work inspired the invention of the patch-alignment method. The patch-alignment method does not use any complex advanced deformable convolution and other techniques, but frees VSR Transformers from the heavy complex alignment module design in a very simple way. Note that the alignment module occupies nearly 1/3 of the parameters of the VRT.
>
> `R1-Q2`: Thanks for the question. We respond to your sub-questions one by one:
>
> First, we do NOT argue that no alignment is better in the case of small motion because the feature after alignment exceeds the attention window. We argue that, if we use alignment, its inaccurate optical flow and interpolation warping method is the source of the performance loss. In the range that the VSR Transformer can handle, no alignment does not introduce the sub-pixel information loss caused by alignment. Thus, even if the objects are within the attention window in both cases, no alignment loses less information, and achieves better performance.
>
> Second, you asked about more evidence. However, the PSNR performance comparison is the most intuitive illustration of our argument. Our experimental design precisely controls the variables, it can reflect the effect of alignment under different motion conditions. Reviewer #1 is encouraged to check the experiments in Table 2 and Table 3. These experiments verify the negative impact of interpolation methods on information utilization and support the above argument.
>
> Third, we respectfully disagree with the reviewer's statement of "direct interactions with its neighbouring $12\times12$ pixels". In a self-attention layer, only the pixels in the $8\times8$ attention window will directly calculate the attention of each other. Although due to the shift-window mechanism, certain pixels will calculate the attention within another $8\times8$ window in the next layer. However, between the pixels of the non-overlapping parts of the two attention windows, the attention is not calculated directly but indirectly via the shift-window transfer mechanism. Beyond the $8\times8$ range, the VSR Transformer cannot directly compute the correlation between two pixels. So in the case of the movement beyond this range, the performance of no alignment drops significantly.

---

> ### Author Response · Authors · 2022-08-05
> **Further discussion with Reviewer CC5y (denoted as R1)**
>
> Dear reviewer CC5y:
>
> We thank you for the precious review time and valuable comments. We have provided corresponding responses and results, which we believe have covered your concerns. We hope to further discuss with you whether or not your concerns have been addressed. Please let us know if you still have any unclear parts of our work.
>
> Best,
>
> Paper 1445 Authors.

---

> > ### Comment · Reviewer_CC5y · 2022-08-08
> > **Good response**
> >
> > Thanks for your detailed response. I would like to raise my rating.

---

> ### Author Response · Authors · 2022-08-08
> **Send call for follow-up discussions with Reviewer R1 (CC5y)**
>
> Dear Reviewer CC5y:
>
> Thanks again for your precious time and valuable comments.
>
> Initially, Reviewer F7UK (denoted as R3) and Reviewer TWT3 (denoted as R4) both thought positively about our work. After rebuttal, Reviewer j86q (denoted as R2) now agrees that our response solves his/her concerns and rates our work acceptable.
>
> We found that you and R2 share some similar concerns. The concerns include several points as follows:
>
> (1) Discussion with previous works, such as VRT, VSRT, and FGST.
>
> (2) Discussion of the FLOPs and runtime.
>
> (3) Other details about the experiments.
>
> For all your concerns, we have provided responses. Are there any deficiencies in our rebuttal? Whether the corresponding responses and results we provide cover your concerns? The discussion period ending date is August 9. Please let us know if you have any unsolved or other concerns.
>
> Thanks,
>
> Paper 1445 Authors.

---

### Author Response · Authors · 2022-08-02
**General Responses to Reviewers and ACs**

Dear Reviewers and ACs:

We sincerely appreciate all the reviews. They give positive and high-quality comments on our paper with a lot of constructive feedback.

We would like to emphasize that our work not only proposes a novel, simple and effective new method called patch alignment. Our work provides extensive analysis of existing alignment methods. We have developed and used a number of novel analytical tools. Our conclusions are insightful and meaningful. We would like the reviewers and ACs to focus on our analysis part. We believe that the contribution of this part to the field is as important or even more important than our new method.

We have updated our draft to incorporate the insightful suggestions of the reviewers:

According to Reviewer 1 and Reviewer 2’s questions, we add a new section 3 in the supplementary material about "The FLOPs and Runtime of the proposed method".

According to Reviewer 1 and Reviewer 2’s concerns, we add a new section 4 in the supplementary material about the special case of VSRT.

Following Reviewer 3’s suggestion, we complete the description of the significance of our work in the conclusion section.

Following Reviewer 4’s suggestion, we add more descriptions of the proposed method. We also adjust the space before and after the captions of the figures. Inspired by Reviewer 4’s comment, we discuss the scope, limitations and possible future work of our work in more detail in section 5 in the supplementary material.

For the other changes, the final performance of our method is presented. We also add the experiment of the PSRT-recurrent method using 30 frames for training.

In the final version, we will improve other minor points of Reviewer 1, Reviewer 2, Reviewer 3, and Reviewer 4. Thank you all for the valuable suggestions.

Thanks,

Paper 1445 Authors.

---

### Meta-Review · Area_Chair_oGLH · 2022-08-26

**Recommendation:** Accept
**Confidence:** Certain

**Metareview:**

This paper re-thinks the role of alignment in video super-resolution based on transformer models. The video alignment is costly which may need manual efforts. This paper proposed several inspiring and counter-intuitive remarks, such as that alignment is unnecessary and may be harmful to the transformer model. The authors presented a new model with only patch alignment instead of the pixel ones and larger window size, which achieves non-trivial improvements over the SOTA methods. Most of the reviewers agreed with the contribution and significance of this paper to the community.

**Award:**

No

---

### Decision · Program_Chairs · 2022-09-14

Accept